# Ageing-induced weakness of mouse NMJs is associated with reduced active zone density, synaptic event kinetics and presynaptic calcium entry

Yizhi Li, Elinor H. Case, Christopher Blanchard, Anna Monteleone, Meera Gandhi, Anousha Jaie 🆔, Yomna Badawi 🆔 and Stephen D. Meriney 🆔

*Department of Neuroscience, University of Pittsburgh, Pittsburgh, PA, USA*

Handling Editors: Katalin Toth & Samuel Young

The peer review history is available in the Supporting information section of this article (https://doi.org/10.1113/JP286735#support-information-section).

**Abstract figure legend** A summary of age-related changes contributing to reduced neurotransmission in mouse neuromuscular junctions (NMJ). At 26 months, NMJs showed potentiation in short-term plasticity compared to 4- or 30-month-old synapses, which is a proxy for reduced probability of release within individual AZs (dashed arrow). Compared to NMJs from 4–26-month-old mice, NMJs from 30-month-old mice showed reduced active zone (AZ) density, a slower vesicle replenishment rate and reduced presynaptic calcium entry following a single action potential stimulus. These factors may contribute to the decline in neurotransmission at NMJs from aged mice.

**Abstract** Ageing has been shown to affect both the structure and function of the neuromuscular junction (NMJ). In our previous study, we documented a biphasic change (first an increase followed by a decrease) in neurotransmission over the ageing time course at male mouse NMJs. Here, we explored several potential mechanisms behind the reduction in presynaptic neurotransmitter release in the later stages of ageing. We found that the active zone (AZ) density significantly decreased in NMJs at 30 months, compared to 4- and 26-month-old mice. Furthermore, the decreased end plate potential (EPP) amplitude in these 30-month-old mice was associated with a significantly longer rise and decay time, and, although miniature end plate potential (mEPPs) amplitude and frequency were unchanged, these events also had a longer rise and decay time. Thirty-month-old NMJs also showed a significant reduction in the vesicle replenishment rate (VRR). Additionally,

**Yizhi (Nick) Li** is currently a Post-Doctoral Associate in the Department of Neuroscience at the University of Pittsburgh. He received his PhD from the Department of Neuroscience at the University of Pittsburgh. He adapted ageing at the neuromuscular junction as his research model to further his research on synaptic plasticity. His goal is to elucidate the role of the neuromuscular junction in age-related muscle weakness and to explore and test novel treatments that target this synapse in ageing.

30–33-month-old NMJs had reduced presynaptic calcium entry following a single presynaptic action potential. Taken together, these transmitter release site changes may explain age-induced reductions in neuromuscular neurotransmission in aged mice and may also lead to the identification of novel therapeutic targets or potential biomarkers for future research in this area.

(Received 14 March 2025; accepted after revision 22 July 2025; first published online 10 August 2025)

**Corresponding author** Stephen D. Meriney: Department of Neuroscience, University of Pittsburgh, Pittsburgh, PA 15260, USA.   Email: meriney@pitt.edu

## Key points

- Neuromuscular junctions (NMJs) from 30-month-old mice, in comparison with 4- and 26-month-old mice, showed altered neurotransmitter release properties; namely, decreased quantal content, decreased end plate potential (EPP) amplitude with prolonged rise and decay time, and prolonged miniature end plate potential (mEPP) rise and decay time.
- The density of transmitter release sites (active zones) was reduced in NMJs from 30-month-old mice compared to 4- and 26-month-old NMJs.
- NMJs from 30-month-old mice showed no change in short-term synaptic plasticity compared to 4-month-old mice, whereas NMJs from 26-month-old mice showed significant synaptic facilitation, and this was restricted to the weakest synapses in this age group.
- NMJs from 30-month-old mice showed no change in readily releasable pool (RRP), but a slower vesicle replenishment rate (VRR) compared to 4-month-old NMJs.
- NMJs from 30–33-month-old mice showed a reduction in presynaptic calcium entry following a single action potential stimulus.

## Introduction

As a result of the burden that age-induced muscle weakness has on our ageing population, it is important to understand the biological mechanisms behind age-related weakness of the neuromuscular system. Our group previously published an article documenting the timeline of age-induced changes at the neuromuscular junction (NMJ) from epitrochleoanconeous (ETA) muscles in male mice, and one of the findings was a biphasic change in neurotransmission during the ageing process: an increase in the early stage of ageing, followed by a decrease in late stage ageing (Li et al., 2023). We are particularly interested in the mechanisms that underlie the decline in neurotransmission at the late stages of ageing because of the potential for the development of treatments or preventative measures for age-related muscle weakness. Therefore, the focus of the present study is on the synaptic mechanisms that underlie reductions in neurotransmission at later aged synapses. Transmitter release from the NMJ occurs from the hundreds of individual release sites (active zones, AZs) that populate each NMJ (Badawi & Nishimune, 2018; Laghaei et al., 2018). As such, we have focused on structural and functional changes within these AZs.

AZs are specialised sites for vesicular exocytosis at the presynaptic membrane (Couteaux & Pecot-Dechavassine, 1970). Electron microscope tomography studies have demonstrated the presence of protein complexes at AZs (Harlow et al., 2001; Harlow et al., 2013; Jung et al., 2018; Nagwaney et al., 2009) located opposite of the postsynaptic junctional folds, where the ACh receptors (AChRs) reside (Chen et al., 2012; Hirokawa & Heuser, 1982). Using immunohistochemistry, it was also demonstrated that these presynaptic AZs at the mouse NMJ can be marked by labelling AZ-specific proteins, such as bassoon, piccolo and the P/Q type calcium ($Ca^{2+}$) channel (Chen et al., 2011, 2012; Nishimune et al., 2004, 2016; Patton et al., 2001). Prior studies focused on the AZ density within NMJs reported 2.3–2.7 AZs per $\mu m^2$ in healthy adult NMJs (Chen et al., 2012; Ellisman et al., 1976; Fukunaga et al., 1982, 1983; Fukuoka et al., 1987; Ruiz et al., 2011). The density and function of NMJ AZs are predicted to have a profound impact on the strength of a synapse.

Another mechanism that could lead to a decline in neurotransmission is a change in presynaptic synaptic vesicle pool sizes or kinetics of trafficking. The contribution of different pools of synaptic vesicles to transmitter release can be estimated using electrophysiology recordings of postsynaptic events during a high-frequency train stimulation of the presynaptic nerve. During a prolonged high-frequency train of stimuli, the

size of the postsynaptic response declines at different rates that are assumed to be reflective of the depletion rates for each pool of vesicles. In particular, the depletion of the docked synaptic vesicles occurs quickly and defines the readily releasable pool (RRP) (Alabi & Tsien, 2012). Once the RRP is depleted after several stimuli, the subsequent transmitter release is now dependent on the vesicles from a 'recycling pool' replenishing the RRP (Elmqvist & Quastel, 1965; Richards et al., 2003). This rate of replenishment of the RRP is termed the vesicle replenishment rate (VRR) (Birks & Macintosh, 1961; Larkman et al., 1991). Any changes in the RRP or VRR are probably important contributors to changes in synaptic function at the NMJ.

Short-term synaptic plasticity is strongly controlled by presynaptic residual $Ca^{2+}$ and vesicle depletion; therefore, this measure can be used to probe mechanisms within the nerve terminal AZs that contribute to the probability of transmitter release at each AZ (Oleskevich et al., 2000; Trommershauser et al., 2003; Zucker & Regehr, 2002). Specifically, we can use this measure to examine synaptic dynamics within AZs at ageing NMJs.

Lastly, presynaptic $Ca^{2+}$ entry during an action potential is crucial for neurotransmitter release, and the magnitude of neurotransmission is tightly controlled by the magnitude of $Ca^{2+}$ influx during presynaptic action potentials (Augustine et al., 1987; Dodge Jr. & Rahamimoff, 1967; Katz & Miledi, 1967, 1970; Zucker & Regehr, 2002). Although it has been previously reported that the fluorescence intensity for immunohistochemically labelled presynaptic voltage-gated calcium channel (VGCC) was significantly reduced in aged NMJs (Nishimune et al., 2016), it is unclear whether presynaptic $Ca^{2+}$ entry levels are altered following a presynaptic action potential at aged NMJs.

Therefore, in the present study, we have investigated the impact of later stages of ageing on the density of AZs, short-term synaptic plasticity, RRP, VRR and presynaptic $Ca^{2+}$ entry during a single presynaptic action potential. Understanding these mechanisms during the ageing of the neuromuscular system not only provides insights into the mechanisms of ageing at the NMJ, but also may lead to novel therapeutic targets or potential biomarkers for future research in this area.

## Methods

### Ethical approval

All animals used in this study were male mice of the C57/BL6 background obtained from Charles River Laboratories (NIA colony of aged mice) (https://www.nia.nih.gov/research/dab/aged-rodent-colonies) and all the animals were housed in traditional mouse cages without physical/environmental enhancements and had access to food and water *ad libitum*. Animal studies were performed according to protocols approved by the University of Pittsburgh Institutional Animal Care and Use Committee (IACUC, Assurance #: D16-00118; Protocol #: 25036287). This research complies with policies of the journal regarding animal experiments.

### Animals

For this study, we have chosen to focus on three time points (4, 26 and 30 months) based on a previously published study (Li et al., 2023) detailing the ageing time course of the male mouse ETA NMJ. These time points represent young adulthood (4 months) and the beginning (26 months) and end (30 months) of the later aged time period. These two specific ages of the later aged time period were chosen to enable us to evaluate synaptic properties over the range of ages during which transmitter release begins to decline and compare with what is observed at young adult neuromuscular synapses.

For most of these studies, we have used surface muscle fibres from the mouse ETA muscle; a thin upper arm muscle that is almost entirely fast-twitch. At the young adult age, surface fibres in the ETA are either type IIb (54% of all fibres) or type IIx (30% of all fast-twitch fibres do not stain for type IIa or IIb antibodies) and the remaining fibres in the interior of ETA are type IIa (16%) (Rogozhin et al., 2008; Villarroel-Campos et al., 2022). For $Ca^{2+}$ imaging studies, we have used a thin scalp muscle that controls the movement of the ears (the levator auris longus; LAL) because $Ca^{2+}$ sensitive dyes reach nerve terminals through the cut end of the nerve faster in this muscle as a result of a short nerve length (similar $Ca^{2+}$ imaging studies in the ETA muscle is more difficult). The LAL is also a predominately fast-twitch muscle containing mostly IIa and IIb fibres (Erzen et al., 2000). Following death of the animal ($CO_2$ inhalation, followed by thoracotomy), the ETA or LAL was dissected and prepared as an *ex vivo* nerve-muscle preparation. Normal Mammalian Ringer solution (NMR; 150 mM NaCl, 5 mM KCl, 11 mM glucose, 10 mM HEPES, 1 mM $MgCl_2$ and 2 mM $CaCl_2$ at pH 7.4) was used to bathe the preparation together with continuous oxygenation.

### Intracellular microelectrode recordings at the mouse NMJ

The ETA muscle nerve was stimulated with a suction electrode at $10\times$ threshold and 1 μM μ-conotoxin GIIIB (Alomone Labs Ltd, Jerusalem, Israel) was added to block action potential-evoked muscle contraction (Hong & Chang, 1989). Borosilicate electrodes with ~40–60 MΩ resistance were filled with 3 M potassium acetate to perform microelectrode recordings. To assess

the magnitude of transmitter release at each NMJ, spontaneous miniature endplate potentials (mEPPs) were collected for 1 min followed by 10 evoked endplate potentials (EPPs) at 0.2 Hz. Then, a 100 Hz train stimulation for 0.5 s was delivered to the muscle nerve, and EPPs were collected to assess short-term synaptic plasticity. The preparation was allowed a 5 min rest between each recording. For some recordings, the same NMJ that was used for intracellular recordings was identified and processed for immunohistochemistry. These specific NMJs were identified later in confocal images of fixed tissue based on a detailed map of surface fibres that was generated during the recordings. Electrophysiological data were corrected for non-linear summation (McLachlan & Martin, 1981). The quantal content (QC) was calculated by dividing the average amplitude of the EPP by the average amplitude of mEPPs recorded from the same synapse. QC was used to estimate the number of vesicles (quanta) released with each presynaptic action potential. Data were collected using an Axoclamp 900A (Molecular Devices, San Jose, CA, USA) and digitised at 10 kHz for data analysis using pClamp 10 (Molecular Devices). During analysis of the electrophysiology data, a 2000 Hz Gaussian low pass filter was for both mEPP and EPP data.

### Readily releasable pool and vesicle replenishment rate analysis

The RRP and VRR were calculated using the data from a 100 Hz train stimulation of the muscle nerve for 0.5 s. For analysis, the cumulative QC was plotted as a function of the stimulus number. A linear extrapolation line was generated using the last 20 stimuli in the train, and the RRP size was estimated by the *y*-intercept, whereas the slope of this line was used to estimate the VRR as previously described (Ruiz et al., 2011; Schneggenburger et al., 1999).

### Immunohistochemistry and confocal imaging

Immunohistochemical staining of proteins at the NMJ was performed after the intracellular recordings from each muscle as described previously (Laghaei et al., 2018). The ETA nerve-muscle preparation was incubated with NMR containing 2 µg mL$^{-1}$ $\alpha$-Bungarotoxin-Alexa 488 (BTX; Thermo Fisher Scientific, Waltham, MA, USA) for 30 min, allowing specific binding to and subsequent visualisation of postsynaptic AChRs. The preparation was then washed with NMR and fixed with 2% paraformaldehyde for 20 min. Permeabilisation and blocking were performed in phosphate-buffered saline (PBS) with 2% bovine serum albumin, 2% goat serum and 0.5% Triton-X 100. The primary antibody was pre-

pared in the same blocking and permeabilisation buffer. Although labelling for the protein bassoon is traditionally used to identify presynaptic AZs, bassoon has been shown to be reduced at aged NMJs (Nishimune et al., 2016). However, in the same study, it was shown that the detection of another AZ specific protein (piccolo) is resistant to the effect of age. In our preliminary studies, we found that the AZ protein RIMBP2 was also resistant to the effects of ageing. Therefore, for measurements of AZ density in this study, we localised the AZ proteins piccolo (dilution 1:1000, guinea pig-anti-piccolo; Synaptic Systems, Goettingen, Germany) and RIMBP2 (dilution 1:1000, rabbit-anti-RIMBP2; Synaptic Systems) following incubation overnight at room temperature. The preparation was then washed and incubated for 4–6 h with secondary antibodies (dilution 1:1000, goat-anti-rabbit-Alexa 647; dilution 1:1000, goat-anti-guinea pig-Alexa 546; Thermo Fisher Scientific) mixed in the blocking and permeabilisation buffer. Afterwards, the preparation was washed in PBS and mounted on slides with the Prolong Gold mounting medium (Thermo Fisher Scientific). A Leica SP8 spectral confocal microscope was (Leica, Wetzlar, Germany) used to image stained NMJs. Confocal image stacks and maximum projection images were obtained using a 0.049–0.076 µm pixel size, 1024 × 512-pixel array, 0.6 µm z-step size and three line averages per frame. An oil immersion objective (HC PL APO CS2 63×/1.40 OIL; Leica) was used to visualise labelled tissue. The lasers used for excitation of secondary antibody labels were 488, 550 and 638 nm. All images were collected at 16-bit resolution. The same NMJs that were previously recorded from were found under the microscope using a map drawn during electrophysiology recordings, and, if these synapses were positioned such that they could be visualised in the mounted tissue, they were imaged for analysis.

### Deconvolution and image analysis

Before image analysis, raw confocal images in the Leica file (.lif) were split into individual channels and saved as TIFF files. NIS-Elements Advanced Research software (Nikon, Tokyo, Japan) was used for deconvolution. The images were deconvolved after 20 iterations. Only the channels that contain the RIMBP2 and piccolo signals were deconvolved. Fiji (Schindelin et al., 2012) was used to analyse the confocal images. For the measurement of AChR area, a manual selection of the area of interest was used following these steps: (1) The channel containing BTX signal was loaded into Fiji. (2) The 'Freehand selection tool' was used to draw the region of interest (ROI). The area within the *en face* BTX signal was selected using the selection tool, excluding the highlights

on the edges of the BTX signal (interpreted to represent a side-view of the edge of a nerve terminal). In the cases where there were highlights on the edges of the BTX-stained NMJ, the ROI was drawn directly adjacent to the highlighted regions. (3) The scale was calibrated by including the pixel size of the image (μm). Then, the area of the AChR signal was measured using the 'Measure' function in Fiji. (4) The ROI was then saved using the ROI manager in Fiji. For the AZ density analysis, the deconvolved images of both RIMBP2 and piccolo were loaded into Fiji. The ROI manager was then used to isolate the selection that overlapped the BTX signal. Finally, the 'Cell Counter' Plugin (https://imagej.net/plugins/cell-counter) was used to manually count the RIMBP2 and piccolo puncta within the ROI. The data presented are the average results from two independent people counting the same images.

### Calcium imaging from the motor nerve terminal

The LAL nerve-muscle preparation (Angaut-Petit & Faille, 1987; Erzen et al., 2000; Ojeda et al., 2020) was isolated and bathed in NMR. This thin muscle was chosen for accessibility of superficially positioned motor nerve terminals and because NMJs are a short distance from muscle nerve entry into the muscle (facilitating loading of $Ca^{2+}$-sensitive dye). Calbryte-520 potassium salt fluorescent dye (#20658; AAT Bioquest, Pleasanton, CA, USA) was loaded into the cut end of the motor nerve as previously described (Luo et al., 2011). The end of the motor nerve was isolated within a Vaseline chamber with the rest of the nerve-muscle preparation against the side of the chamber and bathed in NMR. The end of the nerve inside the chamber was rinsed in deionised water and then immersed in 2 μL of 30 mM Calbryte-520 potassium salt dye mixed in deionised water. The tip of the nerve was then freshly cut in the dye solution to aid in dye loading through the cut end. Then, the Vaseline chamber was covered with a plastic coverslip to prevent evaporation. The preparation was incubated at 25°C for 2 h and then moved to 4°C for one additional hour of incubation. The dye was then removed, and the LAL was rinsed in NMR and pinned in a 35 mm dish over a raised Sylgard platform. Throughout this dye loading procedure, and subsequent imaging, the saline was bubbled with 95% $CO_2$/5% $O_2$. Before imaging, the twitch threshold for nerve stimulation was determined and NMJs were stained using a 30 min incubation in a 4 μg mL$^{-1}$ solution of $\alpha$-bungarotoxin-Alexa 594 (B13423; Invitrogen, Waltham, MA, USA) in NMR. After labelling, the preparation was washed three times in NMR, then incubated in 20 μM tubocurarine chloride hydrate (T2379; Sigma, St Louis, MO, USA) to block any residual nerve-evoked muscle contractions that were not blocked by the $\alpha$-bungarotoxin-Alexa 594. To evoke action potential-triggered $Ca^{2+}$ entry into the motor nerve terminal, the muscle nerve was stimulated using a suction electrode. $Ca^{2+}$ signals were imaged using the XT line-scan mode of a Leica SP8 with a HC APO LUVI 63×/0.90 NA water-immersion objective, at a galvo speed of 700, 2× zoom, 0.172 μm pixel size and 7% laser power. Images were saved as TIFF files and imported into ImageJ for analysis. For each synapse, an ROI was created around the line scan data that covered the motor nerve terminal. The 'plot profile' function was used to average the intensity of each line inside the ROI and provide a plot of the intensity of the $Ca^{2+}$ signal over time. In Excel (Microsoft Corp., Redmond, WA, USA), the baseline fluorescence for each image was determined by averaging the first 100 values before nerve stimulation. Then, the function (raw $F$ – baseline $F$)/baseline $F$ (where raw $F$ is the average fluorescence of each line captured during the experiment and baseline $F$ is the average value of the fluorescence of the first 100 lines before the stimulation) was performed on each data point to normalise the data. Then, the corresponding normalised values from each image were averaged. The resulting average dataset was used to generate the time course of the detected $Ca^{2+}$ signal for each synapse. Twenty values after nerve stimulation were averaged to determine the peak $Ca^{2+}$ signal for each synapse. The data were expressed as $\Delta F/F$ (%) by multiplying the averages of the normalised values by 100.

### Statistical analysis

Statistical analysis was performed using Prism, version 10.2.0 (GraphPad Software Inc., San Diego, CA, USA). Data are presented as the mean ± SD unless otherwise noted. An $\alpha$ of 0.05 was used for all statistical tests.

## Results

We were specifically interested in studying synaptic mechanisms associated with reduced transmitter release at aged NMJs. Therefore, we chose three ages for study: 4, 26 and 30 months. Twenty-six and 30 months represent the beginning and end of the later aged time period based on a previous report on aged NMJs from our laboratory (Li et al., 2023). Here, we explored the relationship between function and structure during late-stage ageing of the NMJs.

### Thirty-month-old NMJs showed altered neurotransmitter release properties

First, we compared transmitter release properties between NMJs at 4-, 26- and 30-month-old mouse NMJs. Using

intracellular recordings of membrane potential, we found that EPP amplitude and QC measured at individual NMJs showed a significant reduction in 30-month-old mice compared to their 4-month and 26-month-old counterparts (EPP means: 4 months = 18.64 ± 7.77 mV; 26 months = 16.28 ± 8.14 mV; 30 months = 10.85 ± 5.12 mV; QC means: 4 months = 80.67 ± 20.92; 26 months = 76.72 ± 21.84; 30 months = 55.89 ± 22.39). By contrast, mEPP amplitude and mEPP frequency showed no significant differences between the three age groups (mEPP amplitude means: 4 months = 0.23 ± 0.09 mV; 26 months = 0.2 ± 0.06 mV; 30 months = 0.24 ± 0.11 mV; mEPP frequency means: 4 months = 5.33 ± 2.6 mEPP $s^{-1}$; 26 months = 4.66 ± 2.37 mEPP $s^{-1}$; 30 months = 4.25 ± 1.9 mEPP $s^{-1}$). mEPP rise time was significantly increased at 30 months compared to 4 and 26 months (means: 4 months = 0.86 ± 0.32 ms; 26 months = 0.94 ± 0.32 ms; 30 months = 1.5 ± 0.66 ms). mEPP decay time was significantly increased at 30 months compared to 4 and 26 months (means: 4 months = 3.5 ± 0.98 ms; 26 months = 3.82 ± 0.84 ms; 30 months = 4.8 ± 1.4 ms). Furthermore, EPPs also showed a significant increase both in rise time from 4 months to 26 months and then from 26 months to 30 months (means: 4 months = 0.86 ± 0.18 ms; 26 months = 0.97 ± 0.12 ms; 30 months = 1.34 ± 0.3 ms) and decay time from 4 months to 26 months and then from 26 months to 30 months (means: 4 months = 4.12 ± 0.93 ms; 26 months = 4.1 ± 0.49 ms; 30 months = 5.5 ± 1.53 ms) (Fig. 1).

### Reduced active zone density in 30-month-old NMJs

Previously, it was shown that the immunohistochemical staining intensity for the traditional AZ marker bassoon was reduced to the degree that AZ density in later aged synapses could not be estimated using this marker, while the staining intensity for another AZ marker (piccolo) remained resistant to the effect of ageing (Nishimune et al., 2016). Using markers for AZs in aged synapses, we could test the hypothesis that AZ density could decrease in later stages of ageing, which could account for reduced transmitter release magnitude (QC). Therefore, we investigated the effect of age on AZ density in neuro-muscular synapses from 4, 26 and 30-month-old mice. When possible, we performed immunohistochemistry from the same NMJs as were used to collect transmitter release data (Fig. 2). Using this approach, we were able to find a subset of the NMJs from which we could record both electrophysiological and immunohistochemical data. For these immunohistochemical experiments, we labelled two presynaptic AZ-specific proteins (piccolo and RIMBP2; another AZ marker we found did not reduce in staining

during ageing) to provide two independent labels for AZ position. The postsynaptic AChRs were labelled with BTX to define the borders of the NMJ in confocal Z-stacks (Fig. 2). We counted the number of piccolo and RIMBP2 puncta as measures of individual AZs within ROI defined by in-focus BTX staining from *en face* regions of the NMJ. We then divided these values by the BTX-stained AChR area to obtain the AZ density per unit micron. We observed no significant differences in the density of piccolo and RIMBP2 puncta within the same NMJ across the three ages (Fig. 2*G*); therefore, we used the average of the two proteins to determine the overall AZ protein density. Compared to 4- and 26-month-old NMJs, 30-month-old NMJs showed a significant decrease in AZ density (AZs/$\mu m^2$) (means: 4 months = 2.04 ± 0.58; 26 months = 2.39 ± 0.54; 30 months = 1.39 ± 0.19) (Fig. 2*H*). Interestingly, we did not detect a significant correlation between AZ protein density and QC (Fig. 2*I*).

### Short-term synaptic plasticity measured from young and aged synapses

Neuromuscular synapses in young adults are thought to vary in size and total AZ number as they develop, grow and adjust to effectively activate their postsynaptic muscle cells (Balice-Gordon & Lichtman, 1990; Hill & Robbins, 1991; Sanes & Lichtman, 1999; Wernig et al., 1984; Wigston, 1989). This normal growth adjustment results in a range of synaptic strengths (QC) for these synapses of different sizes on muscle fibres that vary in diameter and input resistance in young adults (Laghaei et al., 2018). By contrast to this scaling in NMJ size to match postsynaptic muscles, we sought to determine whether there were age-induced changes in short-term synaptic plasticity that might suggest changes in the probability of release per AZ or changes in the $Ca^{2+}$ dependent process that trigger transmitter release. Indeed, a prior report had shown that at later aged synapses (from 29-month-old mice), presynaptic $Ca^{2+}$ channel staining is reduced (Nishimune et al., 2016). QC is a whole synapse measure of the strength of communication at NMJs, but short-term synaptic plasticity is sensitive to local alterations in the AZs within these NMJs. Therefore, we used short-term synaptic plasticity during a 50-stimulus, 100 Hz train as a proxy for alterations in the probability of release or $Ca^{2+}$ sensitive processes within AZs. This measure is sensitive to residual $Ca^{2+}$ between stimuli and AZ vesicle depletion (Zucker & Regehr, 2002). Overall, we found that short-term synaptic plasticity was unchanged when comparing NMJs from young 4-month-old mice with NMJs from later aged 30-month-old mice. Interestingly, NMJs from 26-month-old mice showed significant tetanic potentiation compared to NMJs from 4- and 30-month-old mice (Fig. 3*B* and Table 1).

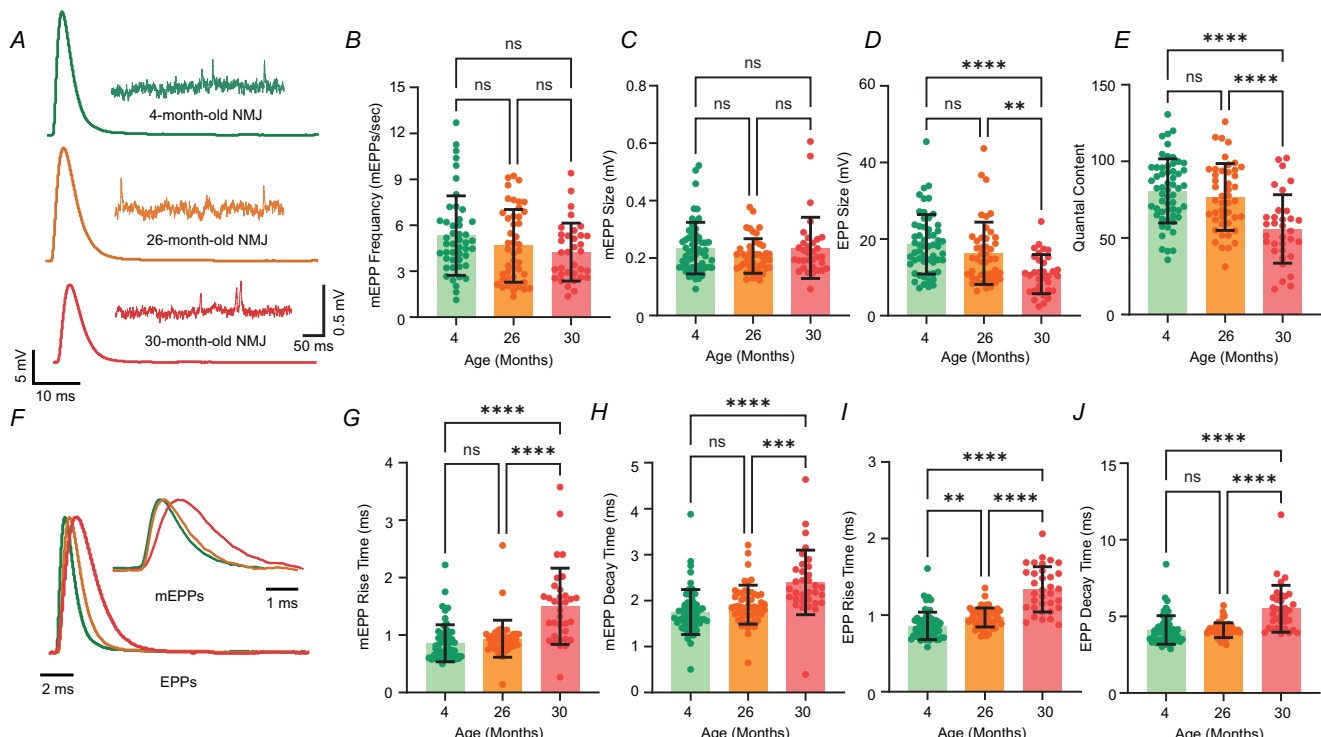

**Figure 1. Transmitter release properties of NMJs from 4-, 26- and 30-month-old mice**

*A*, sample traces of mEPPs and EPPs recorded from 4-, 26- and 30-month-old NMJs. *B*, no significant differences in mEPP frequency recorded from 4, 26 and 30-month-old NMJs ($P = 0.555$ for 4 months *vs*. 26 months; $P = 0.208$ for 4 months *vs*. 30 months; and $P = 0.883$ for 26 months *vs*. 30 months). *C*, no significant differences in mEPP amplitude recorded from 4, 26 and 30-month-old NMJs ($P = 0.377$ for 4 months *vs*. 26 months; $P > 0.999$ for 4 months *vs*. 30 months; and $P = 0.139$ for 26 months *vs*. 30 months). *D*, compared to both 4- and 26-month-old NMJs, 30-month-old NMJs showed a significant reduction in EPP amplitude ($P = 0.383$ for 4 months *vs*. 26 months; ****$P < 0.0001$ for 4 months *vs*. 30 months; and **$P = 0.0087$ for 26 months *vs*. 30 months). *E*, compared to both 4- and 26-month-old NMJs, 30-month-old NMJs showed a significant reduction in QC ($p = 0.77$ for 4 months *vs*. 26 months; ****$P < 0.0001$ for 4 months *vs*. 30 months and ****$P < 0.0001$ for 26 months *vs*. 30 months). *F*, sample traces of mEPPs and EPPs recorded from 4-, 26- and 30-month-old NMJs, scaled to the same peak to show differences in synaptic event kinetics. *G*, compared to both 4- and 26-month-old NMJs, 30-month-old NMJs showed a significant increase in mEPP rise time ($P = 0.626$ for 4 months *vs*. 26 months; ****$P < 0.0001$ for 4 months *vs*. 30 months; and ****$P < 0.0001$ for 26 months *vs*. 30 months). *H*, compared to both 4- and 26-month-old NMJs, 30-month-old NMJs showed a significant increase in mEPP decay time ($P = 0.245$ for 4 months *vs*. 26 months; ****$P < 0.0001$ for 4 months *vs*. 30 months; and ***$P = 0.0002$ for 26 months *vs*. 30 months). *I*, 30-month-old NMJs showed a significant increase in EPP rise time compared to both 4- and 26-month-old synapses, whereas 26-month-old NMJs showed a significant increase in EPP rise time compared to 4-month-old synapses (**$P = 0.0069$ for 4 months *vs*. 26 months; ****$P < 0.0001$ for 4 months *vs*. 30 months; and ****$P < 0.0001$ for 26 months *vs*. 30 months). *J*, compared to both 4- and 26-month-old NMJs, 30-month-old NMJs showed a significant increase in EPP decay time ($P = 0.996$ for 4 months *vs*. 26 months; ****$P < 0.0001$ for 4 months *vs*. 30 months; and ****$P < 0.0001$ for 26 months *vs*. 30 months). The statistical tests performed in (*B*) to (*E*) and (*G*) to (*J*) were a one-way ANOVA with Tukey's *post hoc* test ($n = 57$ for 4-month-old NMJs; $n = 45$ for 26-month-old NMJs; and $n = 33$ for 30-month-old NMJs recorded from three to six animals).

To further examine the increased tetanic potentiation present only at 26-month-old synapses, we investigated the possibility that this potentiation might only be present in a subset of these synapses. There is considerable variability in QC across NMJs from one muscle (Fig. 1*E*), which is consistent with our previous findings (Li et al., 2023). To better understand how synaptic plasticity may be different across this variability in both young adult and aged synapses, we divided the data into single NMJ QC ranges to compare synapses with low, medium and high QC. With a mean QC of about 75 across the datasets taken from both 4- and 26-month-old mice, we set the medium QC range to be $75 \pm 10$ (65–85) ($n = 19$ synapses for 4 months; $n = 13$ synapses for 26 months and $n = 3$ synapses for 30 months). This left synapses with QC lower than 65 in the low QC range ($n = 14$ synapses for 4 months; $n = 15$ synapses for 26 months and $n = 26$ synapses for 30 months),

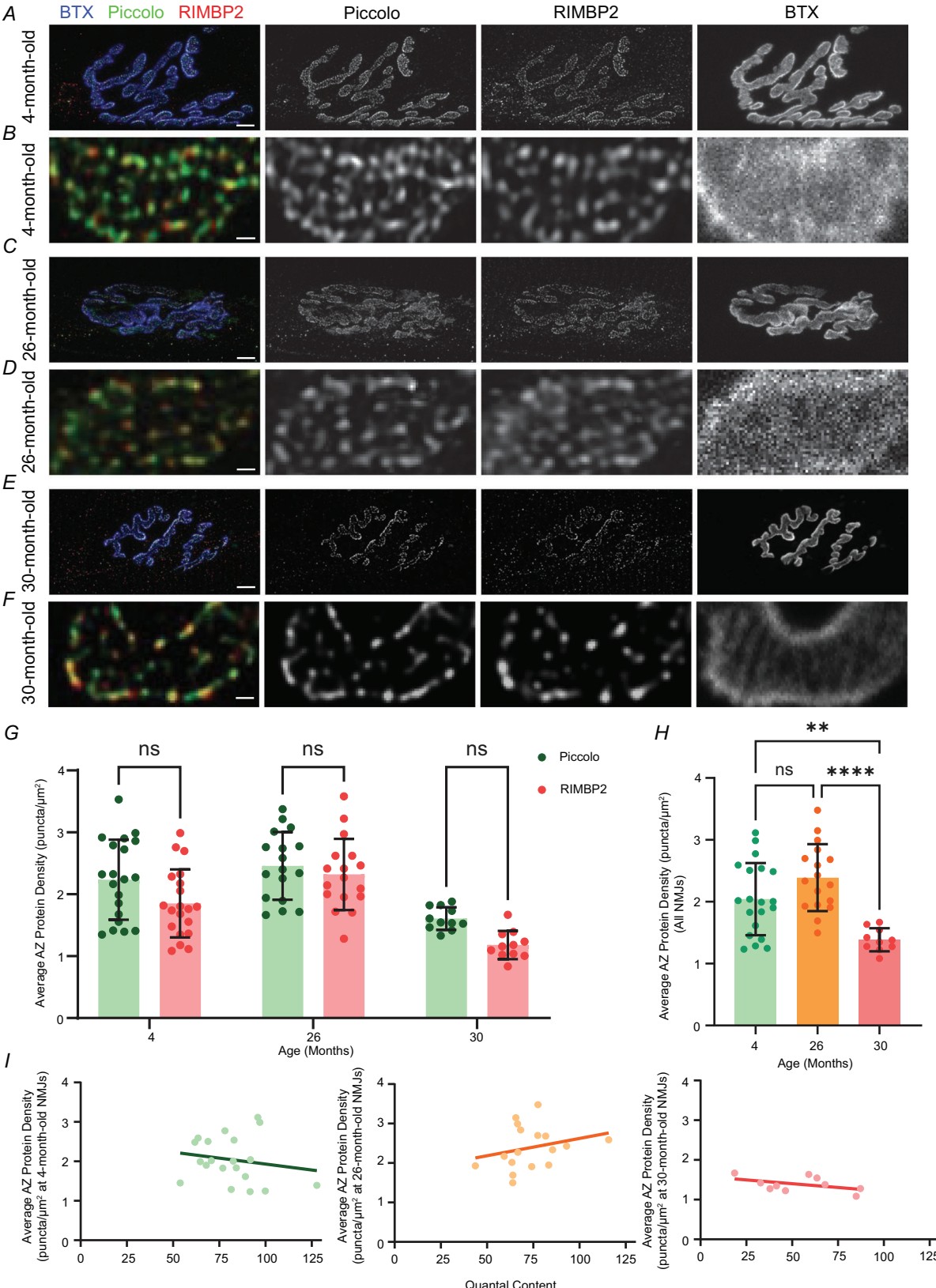

**Figure 2. Active zone density in 4, 26 and 30-month-old NMJs**
Sample confocal maximum projection images showing (left to right) a BTX/piccolo/RIMBP2 composite image, as well as piccolo, RIMBP2 and BTX images from a 4-month-old (*A*), 26-month-old (*C*) and 30-month-old NMJ (*E*); scale

bar = 5 µm. High-magnification sample images of the synapse in (*A*), (*C*) and (*E*) with piccolo/RIMBP2 composite, piccolo, RIMBP2 and BTX images (left to right) from a 4-month-old (*B*), 26-month-old (*D*) and 30-month-old NMJ (*F*); scale bar = 0.5 µm. *G*, comparison of piccolo and RIMBP2 density at 4, 26 and 30-month-old NMJs ($P = 0.0667$ for 4 months; $P = 0.826$ for 26 months; and $P = 0.167$ for 30 months). *H*, 30-month-old synapses at the NMJ showed significantly reduced AZ density compared to both 4-month (**$P = 0.0035$) and 26-month-old synapses (****$P < 0.0001$ and $P = 0.126$ for 4 months *vs.* 26 months) (average of data from RIMBP2 and piccolo labels). *I*, plots demonstrating a lack of significant correlation between average piccolo/RIMBP2 signal density and quantal content at 4- (left), 26- (middle) and 30- (right) month-old NMJs. The statistical tests performed in (*I*) were based on a simple linear regression. The statistical test performed in (*H*) was one-way ANOVA with Tukey's *post hoc* test. Two-way ANOVA was performed in (*G*) ($n = 20$ for 4-month-old NMJs; $n = 17$ for 26-month-old NMJs; and $n = 10$ for 30-month-old NMJs recorded from three to six animals).

whereas synapses with higher than 85 QC were in the high QC range ($n = 24$ synapses for 4 months; $n = 17$ synapses for 26 months and $n = 5$ synapses for 30 months). This method allowed us to compare short-term synaptic plasticity properties based on the synaptic strength of individual synapses. We found that NMJs sorted into low, medium or high QC at the young adult age (4 months) showed no significant differences in short-term synaptic plasticity (Fig. 3*C* and Table 1). This is presumably because differences in QC at this young adult age are due to scaling of NMJ size and AZ number with no changes in the function of individual AZs. By contrast, NMJs with low QC in 26-month-old mice showed a significant increase in tetanic potentiation compared to both low QC synapses in 4- and 30-month-old animals (Fig. 3*D* and Table 1) and compared to medium or high QC synapses from these same 26-month-old mice (Fig. 3*E* and Table 1). As for NMJs in 30-month-old mice, there were no significant differences detected between synapses of different synaptic strengths (Fig. 3*G* and Table 1). These data lead us to hypothesise that there are ageing-induced changes in presynaptic mechanisms that control the probability of release within AZs of only weak synapses at 26 months (an earlier phase of later ageing). These changes are not present in mice that are 4 months old, or in synapses taken from later aged 30-month-old mice.

### Changes in the RRP and the VRR at young and aged synapses

Within synapses, three pools of synaptic vesicles have been defined: readily releasable, recycling and storage (Alabi & Tsien, 2012). The probability of release at synapses can be influenced by the number of docked synaptic vesicles that make up the RRP (Dobrunz & Stevens, 1997; Murthy et al., 2001). The number of vesicles in the RRP, coupled with the release probability, governs the QC (Allen & Stevens, 1994; Stevens & Wang, 1994). Furthermore, at CNS synapses, one mechanism that contributes to synaptic plasticity is a change in synaptic vesicles pools (Murthy et al., 2001), and selective effects on the RRP can be an effective mechanism to alter the probability of release (Goda & Stevens, 1998). Another vesicle pool property that can affect transmitter release is the rate at which synaptic vesicles are replenished

at the release site during repeated use (i.e. VRR). A slower VRR can reduce the number of synaptic vesicles that are available for transmitter release.

At NMJs, the size of the synapse (number of AZs) creates more release sites that can accommodate more docked vesicles that make up the RRP. We estimated the RRP at 4-, 26- and 30-month-old NMJs by measuring the cumulative QC as a function of the stimulus number during a 100 Hz train of nerve stimulation for 30 s (Fig. 4*A*). The resulting plot of cumulative QC *vs.* stimulus number can be used to estimate the RRP and VRR (Fig. 4*B*). A linear function fit to the last 20 stimuli in the train generates the slope of the relationship (an estimate for VRR) and the *y*-intercept (an estimate for RRP). Overall, we found a significant decrease in RRP in 30-month-old synapses compared to 26-month-old synapses (Fig. 4*C*), whereas VRR showed significant reduction in 30-month-old NMJs compared to both 4- and 26-month-old counterparts (Fig. 4*D*). In addition, we found that there was a significant correlation between RRP and QC at 4 months (Fig. 4*E*), 26 months (Fig. 4*G*) and 30 months (Fig. 4*I*). A significant correlation was also found between VRR and QC at 4 months (Fig. 4*F*), 26 months (Fig. 4*H*) and 30 months (Fig. 4*J*).

### Reduced calcium entry during evoked release

Neurotransmitter release is triggered by $Ca^{2+}$ entry following a presynaptic action potential (Augustine et al., 1987; Dodge Jr. & Rahamimoff, 1967; Katz & Miledi, 1967, 1970; Zucker & Regehr, 2002). Given that we have observed a reduction in QC together with a reduction in AZ density at 30 months for NMJs, we tested whether presynaptic $Ca^{2+}$ entry was also reduced in later aged NMJs. Here, we loaded Calbryte-520 into motor nerve endings within LAL muscle-nerve preparations to measure the presynaptic $Ca^{2+}$ entry following a single presynaptic action potential at young (4 months) (Fig. 5*A–C*) *vs.* later aged (30–33 months) (Fig. 5*D–F*) NMJs. Compared to the young NMJs (mean = $52.75 \pm 11.9$), aged NMJs (mean = $34.74 \pm 20.94$) showed significantly reduced increases in $Ca^{2+}$ fluorescence intensity after a single action potential stimulation (Fig. 5*I–K*).

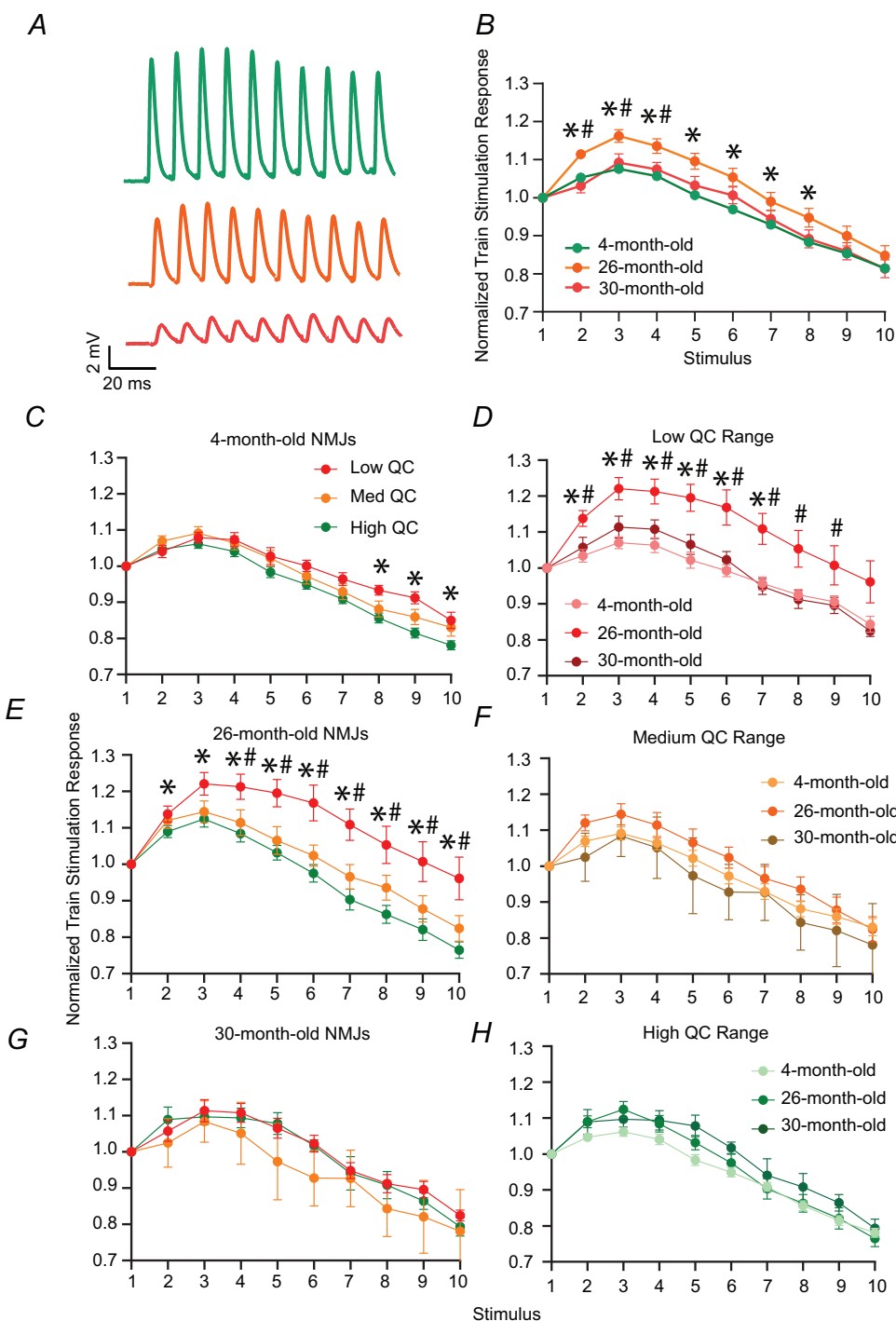

**Figure 3. Short-term synaptic plasticity at 4-, 26- and 30-month-old NMJs**

*A*, sample traces of the first 10 responses during a 100 Hz train stimulation of the muscle nerve for 0.5 s. The stimulus artifacts have been truncated for clarity. *B*, plot of the first 10 stimuli from a 100 Hz train stimulation from 4-, 26- or 30-month-old synapses. *Significance between 4- and 26-month-old NMJs; #significance between 26- and 30-month-old NMJs. Plot of the first 10 stimuli from a 100 Hz train stimulation from 4-month-old (*C*), 26-month-old (*E*) and 30-month-old (*G*) synapses in different QC ranges (red, low QC; orange, medium QC; green, high QC). *Significance between low and high QC NMJs. Plot of the first 10 stimuli from a 100 Hz train stimulation from 4-, 26- and 30-month-old synapses in the low QC range (*D*), medium QC range (*F*) and high QC range (*H*). *Significance between 4- and 26-month-old NMJs; #significance between 26- and 30-month-old NMJs. All statistical tests performed were a one-way ANOVA with Tukey's *post hoc* test (*n* = 57 for 4-month-old NMJs; *n* = 45 for 26-month-old NMJs; and *n* = 33 for 30-month-old NMJs recorded from three to six animals).

**Table 1. Mean ± SD and statistics from short-term synaptic plasticity.**

| Panel in Fig. 3 | Mean ± SD | | | Comparisons (one-way ANOVA with Tukey's *post hoc* test) | | |
|---|---|---|---|---|---|---|
| **B** | 4-month-old | 26-month-old | 30-month-old | 4 *vs.* 26 | 4 *vs.* 30 | 26 *vs.* 30 |
| Stimulus 2 | 1.053 ± 0.0546 | 1.114 ± 0.0793 | 1.038 ± 0.110 | 0.0005 | 0.676 | 0.0001 |
| Stimulus 3 | 1.076 ± 0.0670 | 1.162 ± 0.0112 | 1.085 ± 0.146 | 0.0002 | 0.913 | 0.0051 |
| Stimulus 4 | 1.057 ± 0.0756 | 1.136 ± 0.129 | 1.073 ± 0.109 | 0.0006 | 0.735 | 0.0248 |
| Stimulus 5 | 1.007 ± 0.0867 | 1.096 ± 0.141 | 1.042 ± 0.135 | 0.0008 | 0.36 | 0.122 |
| Stimulus 6 | 0.970 ± 0.0756 | 1.054 ± 0.159 | 1.015 ± 0.125 | 0.0019 | 0.193 | 0.347 |
| Stimulus 7 | 0.929 ± 0.0782 | 0.999 ± 0.161 | 0.955 ± 0.134 | 0.0425 | 0.605 | 0.441 |
| Stimulus 8 | 0.883 ± 0.0807 | 0.947 ± 0.164 | 0.896 ± 0.136 | 0.0344 | 0.886 | 0.185 |
| Stimulus 9 | 0.854 ± 0.0832 | 0.899 ± 0.176 | 0.868 ± 0.132 | 0.195 | 0.868 | 0.552 |
| Stimulus 10 | 0.814 ± 0.0879 | 0.848 ± 0.176 | 0.828 ± 0.140 | 0.438 | 0.894 | 0.793 |
| **C** | Low | Medium | High | Low *vs.* medium | Low *vs.* high | Medium *vs.* high |
| Stimulus 2 | 1.041 ± 0.0640 | 1.069 ± 0.0629 | 1.047 ± 0.0386 | 0.306 | 0.947 | 0.369 |
| Stimulus 3 | 1.079 ± 0.0560 | 1.091 ± 0.0795 | 1.062 ± 0.0618 | 0.861 | 0.739 | 0.342 |
| Stimulus 4 | 1.073 ± 0.0729 | 1.064 ± 0.0822 | 1.041 ± 0.0718 | 0.938 | 0.409 | 0.57 |
| Stimulus 5 | 1.028 ± 0.0879 | 1.022 ± 0.0968 | 0.983 ± 0.0745 | 0.98 | 0.281 | 0.314 |
| Stimulus 6 | 1.001 ± 0.0575 | 0.972 ± 0.0944 | 0.949 ± 0.0635 | 0.518 | 0.106 | 0.579 |
| Stimulus 7 | 0.965 ± 0.066 | 0.929 ± 0.0972 | 0.909 ± 0.0618 | 0.394 | 0.0868 | 0.665 |
| Stimulus 8 | 0.933 ± 0.0515 | 0.881 ± 0.098 | 0.856 ± 0.0674 | 0.135 | 0.0108 | 0.54 |
| Stimulus 9 | 0.912 ± 0.0625 | 0.859 ± 0.0919 | 0.815 ± 0.0658 | 0.12 | 0.0008 | 0.138 |
| Stimulus 10 | 0.85 ± 0.0849 | 0.83 ± 0.106 | 0.781 ± 0.0618 | 0.788 | 0.0452 | 0.14 |
| **D** | 4-month-old | 26-month-old | 30-month-old | 4 *vs.* 26 | 4 *vs.* 30 | 26 *vs.* 30 |
| Stimulus 2 | 1.041 ± 0.064 | 1.144 ± 0.0831 | 1.03 ± 0.116 | 0.014 | 0.985 | 0.0009 |
| Stimulus 3 | 1.079 ± 0.056 | 1.22 ± 0.121 | 1.083 ± 0.164 | 0.0103 | 0.999 | 0.003 |
| Stimulus 4 | 1.073 ± 0.0729 | 1.218 ± 0.128 | 1.072 ± 0.114 | 0.0016 | >0.9999 | 0.0002 |
| Stimulus 5 | 1.028 ± 0.0879 | 1.196 ± 0.145 | 1.043 ± 0.141 | 0.0019 | 0.981 | 0.001 |
| Stimulus 6 | 1.001 ± 0.0575 | 1.161 ± 0.18 | 1.025 ± 0.134 | 0.004 | 0.94 | 0.0046 |
| Stimulus 7 | 0.965 ± 0.066 | 1.102 ± 0.167 | 0.961 ± 0.142 | 0.0184 | 0.999 | 0.0034 |
| Stimulus 8 | 0.933 ± 0.0515 | 1.056 ± 0.185 | 0.9 ± 0.147 | 0.055 | 0.876 | 0.0015 |
| Stimulus 9 | 0.912 ± 0.0625 | 1.015 ± 0.196 | 0.874 ± 0.141 | 0.149 | 0.827 | 0.0055 |
| Stimulus 10 | 0.85 ± 0.0849 | 0.957 ± 0.22 | 0.84 ± 0.147 | 0.186 | 0.997 | 0.0545 |
| **E** | Low | Medium | High | Low *vs.* medium | Low *vs.* high | Medium *vs.* high |
| Stimulus 2 | 1.144 ± 0.0831 | 1.128 ± 0.0506 | 1.068 ± 0.0761 | 0.818 | 0.0136 | 0.104 |
| Stimulus 3 | 1.22 ± 0.121 | 1.144 ± 0.078 | 1.053 ± 0.0847 | 0.127 | 0.0082 | 0.625 |
| Stimulus 4 | 1.218 ± 0.128 | 1.115 ± 0.104 | 1.073 ± 0.109 | 0.0391 | 0.0003 | 0.315 |
| Stimulus 5 | 1.196 ± 0.145 | 1.058 ± 0.0916 | 1.006 ± 0.0862 | 0.0068 | <0.0001 | 0.482 |
| Stimulus 6 | 1.161 ± 0.0756 | 1.003 ± 0.0816 | 0.965 ± 0.0988 | 0.0087 | 0.0005 | 0.749 |
| Stimulus 7 | 1.102 ± 0.167 | 0.954 ± 0.08 | 0.884 ± 0.115 | 0.0116 | <0.0001 | 0.371 |
| Stimulus 8 | 1.056 ± 0.185 | 0.902 ± 0.1 | 0.853 ± 0.0936 | 0.0139 | 0.0004 | 0.638 |
| Stimulus 9 | 1.015 ± 0.196 | 0.858 ± 0.0939 | 0.794 ± 0.114 | 0.0196 | 0.0003 | 0.506 |
| Stimulus 10 | 0.957 ± 0.220 | 0.799 ± 0.0867 | 0.755 ± 0.0811 | 0.0242 | 0.0016 | 0.741 |

*(Continued)*

**Table 1. (Continued)**

| Panel in Fig. 3 | Mean ± SD | | | Comparisons (one-way ANOVA with Tukey's *post hoc* test) | | |
|---|---|---|---|---|---|---|
| **F** | 4-month-old | 26-month-old | 30-month-old | 4 *vs.* 26 | 4 *vs.* 30 | 26 *vs.* 30 |
| Stimulus 2 | 1.069 ± 0.0629 | 1.128 ± 0.0506 | 1.025 ± 0.117 | 0.312 | 0.859 | 0.306 |
| Stimulus 3 | 1.091 ± 0.0795 | 1.144 ± 0.078 | 1.084 ± 0.0986 | 0.548 | 0.999 | 0.822 |
| Stimulus 4 | 1.064 ± 0.0822 | 1.115 ± 0.104 | 1.051 ± 0.148 | 0.588 | 0.998 | 0.801 |
| Stimulus 5 | 1.022 ± 0.0968 | 1.058 ± 0.0916 | 0.973 ± 0.183 | 0.851 | 0.918 | 0.704 |
| Stimulus 6 | 0.972 ± 0.0944 | 1.003 ± 0.0816 | 0.928 ± 0.134 | 0.856 | 0.908 | 0.691 |
| Stimulus 7 | 0.929 ± 0.0972 | 0.954 ± 0.08 | 0.927 ± 0.135 | 0.922 | >0.9999 | 0.979 |
| Stimulus 8 | 0.881 ± 0.098 | 0.902 ± 0.1 | 0.843 ± 0.134 | 0.959 | 0.953 | 0.859 |
| Stimulus 9 | 0.859 ± 0.0919 | 0.899 ± 0.176 | 0.868 ± 0.132 | >0.9999 | 0.929 | 0.94 |
| Stimulus 10 | 0.83 ± 0.106 | 0.799 ± 0.0867 | 0.781 ± 0.199 | 0.867 | 0.882 | 0.993 |
| **G** | Low | Medium | High | Low *vs.* medium | Low *vs.* high | Medium *vs.* high |
| Stimulus 2 | 1.03 ± 0.116 | 1.025 ± 0.117 | 1.089 ± 0.0772 | 0.851 | 0.995 | 0.922 |
| Stimulus 3 | 1.083 ± 0.164 | 1.084 ± 0.0986 | 1.097 ± 0.0487 | 0.258 | 0.532 | 0.95 |
| Stimulus 4 | 1.072 ± 0.114 | 1.051 ± 0.148 | 1.094 ± 0.0605 | 0.0156 | 0.47 | 0.376 |
| Stimulus 5 | 1.043 ± 0.141 | 0.973 ± 0.183 | 1.078 ± 0.0676 | 0.017 | 0.449 | 0.411 |
| Stimulus 6 | 1.025 ± 0.134 | 0.928 ± 0.134 | 1.017 ± 0.0356 | 0.0355 | 0.228 | 0.843 |
| Stimulus 7 | 0.961 ± 0.142 | 0.927 ± 0.135 | 0.941 ± 0.103 | 0.0111 | 0.207 | 0.641 |
| Stimulus 8 | 0.9 ± 0.147 | 0.843 ± 0.134 | 0.908 ± 0.0843 | 0.0002 | 0.0173 | 0.662 |
| Stimulus 9 | 0.884 ± 0.141 | 0.821 ± 0.174 | 0.864 ± 0.0513 | 0.0002 | 0.107 | 0.187 |
| Stimulus 10 | 0.84 ± 0.147 | 0.781 ± 0.199 | 0.793 ± 0.0575 | <0.0001 | 0.0281 | 0.27 |
| **H** | 4-month-old | 26-month-old | 30-month-old | 4 *vs.* 26 | 4 *vs.* 30 | 26 *vs.* 30 |
| Stimulus 2 | 1.047 ± 0.0386 | 1.068 ± 0.0761 | 1.089 ± 0.0772 | 0.716 | 0.502 | 0.911 |
| Stimulus 3 | 1.062 ± 0.0618 | 1.107 ± 0.0952 | 1.097 ± 0.0487 | 0.327 | 0.812 | 0.995 |
| Stimulus 4 | 1.041 ± 0.0718 | 1.053 ± 0.0847 | 1.094 ± 0.0605 | 0.965 | 0.553 | 0.774 |
| Stimulus 5 | 0.983 ± 0.0745 | 1.006 ± 0.0862 | 1.078 ± 0.0686 | 0.802 | 0.0758 | 0.295 |
| Stimulus 6 | 0.949 ± 0.0635 | 0.965 ± 0.0988 | 1.017 ± 0.0356 | 0.913 | 0.262 | 0.54 |
| Stimulus 7 | 0.909 ± 0.0618 | 0.884 ± 0.115 | 0.941 ± 0.103 | 0.858 | 0.902 | 0.655 |
| Stimulus 8 | 0.856 ± 0.0674 | 0.853 ± 0.0936 | 0.908 ± 0.0843 | 0.999 | 0.599 | 0.596 |
| Stimulus 9 | 0.815 ± 0.0658 | 0.794 ± 0.114 | 0.864 ± 0.0513 | 0.88 | 0.642 | 0.39 |
| Stimulus 10 | 0.781 ± 0.0618 | 0.755 ± 0.0811 | 0.793 ± 0.0575 | 0.724 | 0.985 | 0.752 |

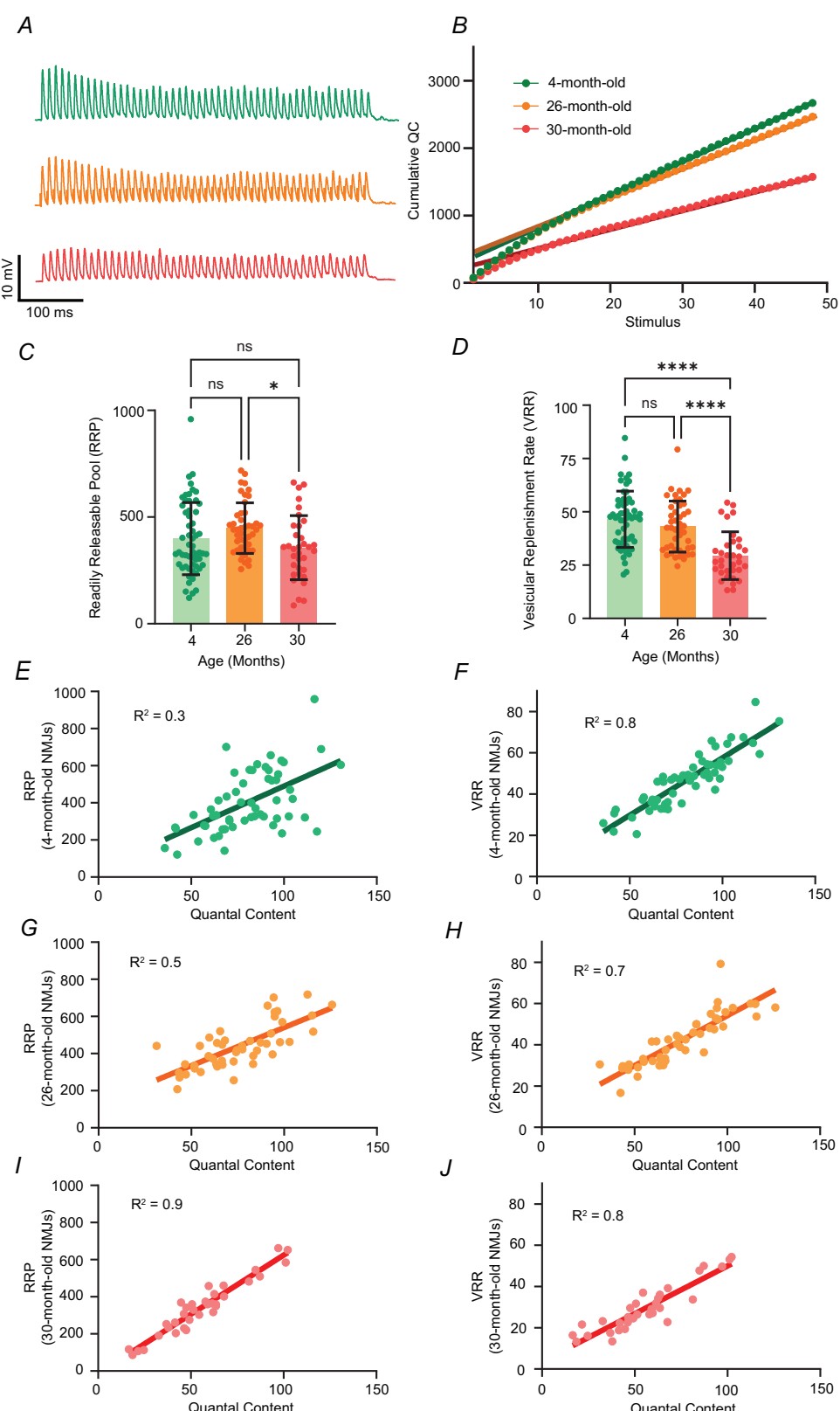

**Figure 4. RRP and VRR are significantly reduced at NMJs in 30-month-old mice**

*A*, sample traces of the postsynaptic EPPs during a 100 Hz train stimulation from 4- (top green)-, 26 (middle yellow)- and 30 (bottom red)-month-old NMJs. *B*, cumulative QC amplitude as a function of the stimulus number

from NMJs at 4, 26 and 30 months (linear regression line is drawn based on the last 20 points in the plot). *C*, 30-month-old NMJs showed a significant reduction in RRP compared to 26-month-old NMJs (*$P$ = 0.0169; $P$ = 0.315 for 4 months *vs.* 26 months and $P$ = 0.51 for 4 months *vs.* 30 months). *D*, 30-month-old NMJs showed a significant reduction in VRR compared to both 4 months (****$p$ <0.0001) and 26 months (****$P < 0.0001$ and $P$ = 0.449 for 4 months *vs.* 26 months). QC and RRP are significantly correlated in NMJs of 4-month-old (*E*), 26-month-old (*G*) and 30-month-old (*I*) mice. QC and VRR are significantly correlated in NMJs of 4-month-old (*F*), 26-month-old (*H*) and 30-month-old (*J*) mice. Statistical tests performed in (*C*) and (*D*) were a one-way ANOVA with Tukey's *post hoc* test. Statistical tests performed in (*E*) to (*J*) were based on a simple linear regression ($n$ = 57 for 4-month-old NMJs; $n$ = 45 for 26-month-old NMJs; and $n$ = 33 for 30-month-old NMJs recorded from three to six animals).

## Discussion

In the present study, we investigated the synaptic mechanisms that might underlie decreases in neurotransmission at later ageing time points and chose two ages (26 months and 30 months) to represent the beginning and end of the later aged phase. We found that transmitter release sites within NMJs of 26-month-old mice showed mostly unchanged synaptic release properties. However, at this early phase of ageing, we did find that short-term synaptic plasticity resulted in significant tetanic potentiation. We hypothesise that this 26-month-old time period might represent the end of a homeostatic compensation period that exists in early ageing that was identified in our earlier study (Li et al., 2023), which occurs before transmitter deficits are observed at 30 months (later ageing). In other words, we hypothesise that ageing NMJs go through an early ageing phase that is characterised by compensatory homeostatic plasticity as the system attempts to correct for age-induced changes at the NMJ, and a later ageing phase where functional and structural deficits manifest and lead to reduced transmitter release. Here, we have focused on the deficits present in the later ageing phase and acknowledge that homeostatic mechanisms in the early ageing phase will require additional studies that are beyond the scope of this report. By focusing this study on synaptic mechanisms of later age-induced weakness at the NMJ, we can focus on biological events here, rather than the chronological focus of the previous study from our laboratory (Li et al., 2023). We can also identify potential later ageing biomarkers and/or potential targets for future therapeutic developments. In particular, we started by examining changes in AZ density, short-term plasticity, the RRP size and the VRR. We found that, at 30 months (a later ageing stage), neuromuscular synapses displayed a variety of changes that included a decreased AZ density, an increase in rise and decay time for both mEPPs and EPPs, and a decrease in the VRR. Then, we examined the magnitude of $Ca^{2+}$ entry into the nerve terminal after action potential stimulation and found that this was reduced in later ageing (30–33 months). These changes were not observed in synapses at an early stage of ageing (26 months). Therefore, we conclude that a significant reduction in the AZ density, prolonged rise time of both spontaneous and evoked events, lower VRR, and reduced $Ca^{2+}$ entry could be underlying mechanisms that contribute to the decline in neurotransmission at later stages of ageing.

These observations also add to our growing understanding of the ageing time course. It appears that ageing neuromuscular synapses go through an initial phase where postsynaptic receptors become fragmented (Li et al., 2011, 2023; Prakash & Sieck, 1998; Valdez et al., 2012; Willadt et al., 2016) and an early ageing phase where potential presynaptic homeostatic compensation takes place (Fahim, 1997; Jacob & Robbins, 1990; Li et al., 2023). This present study builds on our previous report (Li et al., 2023) detailing a time course of age-induced structural and functional deficits at the NMJ by further investigating mechanisms responsible for NMJ weakness and reduced neurotransmission in the later ageing phase.

### Implications of an increase in rise and decay time of both spontaneous and evoked events at aged NMJs

Associated with the age-induced reductions in EPP amplitude (Fig. 1*D*) and QC (Fig. 1*E*), NMJs from 30-month-old mice showed significantly increased rise and decay times for both mEPPs (Fig. 1*G* and *H*) and EPPs (Fig. 1*I* and *J*). Because the spatial alignment of the presynaptic release site with the postsynaptic junctional folds that concentrate AChRs are critical for efficient neurotransmission, we hypothesise that changes in the rise and decay times at aged NMJs might be a consequence of misalignment of presynaptic release sites with postsynaptic receptors. One group of proteins that could lead to this misalignment is the laminin protein family, which has been shown to act as an extracellular organiser to aid in the alignment of the presynaptic release sites and postsynaptic receptors (Patton et al., 1997). Among these laminin proteins, laminin $\alpha4$ and laminin $\beta2$ are of particular interest in this context. First, a study using the laminin $\alpha4$ knockout animals showed disrupted alignment of presynaptic AZs with the postsynaptic junctional folds at the NMJ (Patton et al., 2001). Because the expression of laminin $\alpha4$ at the NMJ is reduced during the ageing process (Lee et al., 2017) and laminin $\alpha4$ knockout mice display increased mEPP decay time and decreased AZ density, it is possible that

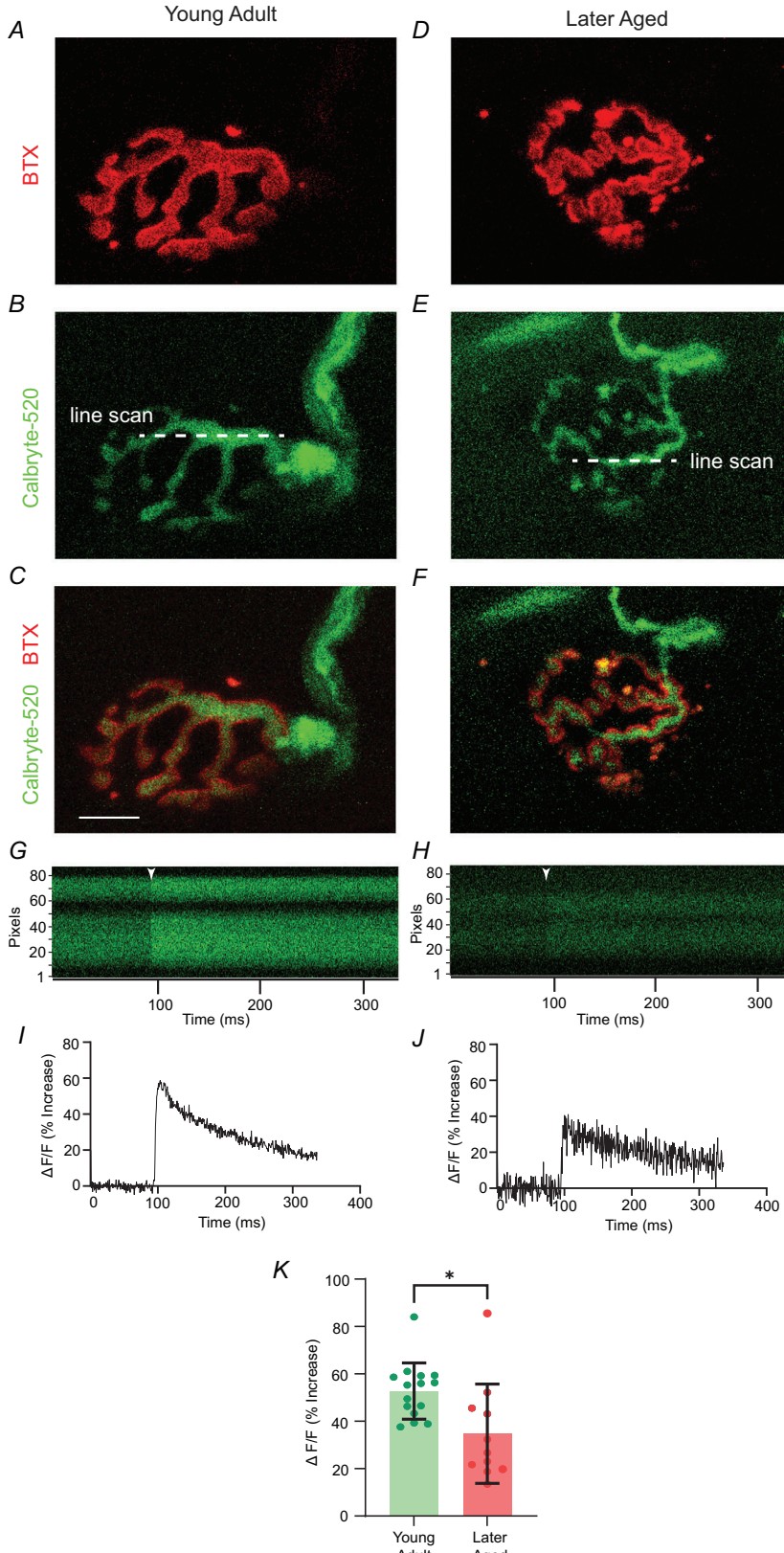

**Figure 5. Calcium entry is significantly reduced in NMJs at 30–33 months**

*A–C*, sample images of a 4-month-old NMJ stained for AChRs (*A*, labelled with BTX), innervating motor nerve and presynaptic nerve terminal (*B*, loaded with Calbryte-520) and the overlay of the two signals (*C*). *D–F*, sample images of a 30-month-old NMJ, stained for AChRs (*D*, labelled with BTX), innervating motor nerve and presynaptic nerve terminal (*E*, loaded with Calbryte-520) and the overlay of the two signals (*F*). *G* and *H*, sample images of Calbryte-520 line scan signal intensity taken from a 4-month-old *G*) and a 30-month-old (*H*) NMJs during a recording session. Arrow heads show the time at which the action potential stimulation was delivered to the motor nerve. *I* and *J*, representative plots of the time course of Ca$^{2+}$-sensitive fluorescence signal over time during the stimulus protocol from a 4-month-old (*I*) and a 30-month-old (*J*) NMJ. *K*, comparison of Calbryte-520 peak signal intensity following the action potential stimulation for both 4-month-old (young adult) and 30–33-month-old (later aged) NMJs. Asterisks indicate significance by an unpaired *t* test (*$P = 0.00102$; $n = 15$ synapses for young adult and $n = 11$ synapses for later aged from three to 12 animals).

 Y. Li and others 

an age-induced loss of laminin $\alpha 4$ may contribute to misalignment of pre- and postsynaptic specialisations and slowed synaptic event kinetics (due to a longer distance for transmitter diffusion). In addition, mutant NMJs lacking laminin $\beta 2$ were reported to have reduced AZ number, lowered QC and EPP amplitude, as well as a trend for increased rise time for both evoked and spontaneous events (Knight et al., 2003; Noakes et al., 1995), consistent with some of the findings in this study. In addition, blocking the interaction between laminin $\beta 2$ and VGCC *in vivo* also reduced the AZ number in wild-type mouse NMJs because laminin $\beta 2$ directly organises AZs through its interaction with VGCCs (Nishimune et al., 2004). Therefore, taken together, it is possible that ageing could disrupt the density and distribution of laminin $\beta 2$ and/or laminin $\alpha 4$ at the NMJ, resulting in the functional deficits we observed in this study. It is also possible that these functional changes in presynaptic release properties might be associated with changes in postsynaptic AChR function, density and/or distribution at aged NMJs, although these potential issues could not be addressed in the present study. Additional studies of synaptic laminin subunits (and potentially other synaptic cleft proteins) at later ageing stages will be required to better understand the link between these extracellular proteins and functional presynaptic changes at aged NMJs.

### A reduction in AZ density at aged NMJs

In the present study, we wanted to associate structure with function and ask if structural changes at the AZ were correlated with changes in transmitter release that we observed in our previous study (Li et al., 2023). To achieve this goal, we quantified the effect of ageing on AZ density at two time points (26 and 30 months) during later ageing phases. We used antibodies against the AZ specific proteins piccolo and RIMBP2 to visualise the location of the AZs at mouse NMJs. Using these fluorescence signals, we were able to calculate the density of AZs. We detected a significant reduction in AZ density only at NMJs from 30-month-old mice compared to NMJs from 4- to 26-month-old mice (Fig. 2*H*). Our counts of AZ density for 4- and 26-month-old synapses were consistent with prior reports: we counted 2.04 AZ per $\mu m^2$ for 4-month-old NMJs and 2.39 AZ per $\mu m^2$ for 26-month-old NMJs, which is consistent with other confocal studies reporting AZ densities to be $\sim 2.3$ AZs per $\mu m^2$ (Chen et al., 2012; Ruiz et al., 2011) and electron microscopy studies reporting a density 2.4–2.7 AZs per $\mu m^2$ (Ellisman et al., 1976; Fukunaga et al., 1982, 1983; Fukuoka et al., 1987). At 30 months, AZ density was significantly reduced to 1.39 AZs per $\mu m^2$. With fewer AZs to participate following a presynaptic action potential, a reduction in transmitter release would be expected,

which we reported in our electrophysiological recordings (Fig. 1*E*). Our results suggest that AZ density is potentially a major contributor to age-related reductions in neurotransmission at 30 months. We observed a large variability in AZ density at 4 and 26 months (Fig. 2*G* and *H*). However, unlike 30-month-old NMJs, synapses with lower QC at 4 and 26 months are not associated with lower AZ density. These data lead us to conclude that differences in synaptic strength at these younger ages may simply be a result of the scaling of NMJ size and AZ number on muscle fibres of different sizes (and input resistance). We propose that low QC NMJs from 4-or 26-month-old mice are probably attributed to smaller NMJs onto smaller muscle fibres, which would create motor nerve terminals with weaker synaptic strength (Harris & Ribchester, 1979; Jones et al., 2016; Ribchester et al., 2004), whereas, at 30 months, synapses display structural deficits in their transmitter release sites that weaken them in a pathological manner.

### Implications of alterations in short-term synaptic plasticity at aged synapses

In our short-term synaptic plasticity analysis, we detected an increase in tetanic potentiation only at NMJs from 26-month-old mice. This enhanced tetanic potentiation could be interpreted as indicative of a lower probability of release or altered $Ca^{2+}$ dynamics at AZs. Because we only detected this change at the weakest NMJs (low QC group) in these 26-month-old mice, we hypothesise that these weaker synapses are undergoing age-induced changes in the function of these AZs. Perhaps the enhanced tetanic potentiation is a result of compensatory mechanisms aimed at correcting reduced function in this early ageing phase. Interestingly, this change in short-term synaptic plasticity was not seen at NMJs later in ageing (30-month-old mice). Given the reduction of AZ density we observed at 30 months, it is possible that the weak AZs in the low QC group of NMJs from 26-month-old mice were eliminated as the NMJ continued to age. It is not known how the density and distribution of AZs might be regulated, but it is possible that AZs with a low probability of release in the early ageing time period might be marked for elimination.

It is also interesting to speculate on the mechanisms that might lead to strongly enhanced tetanic potentiation at weaker synapses only in the early ageing time period (26 months). The probability of release at AZs is often related to the magnitude of $Ca^{2+}$ influx into the presynaptic nerve terminal (Dittrich et al., 2018; Homan et al., 2018; Laghaei & Meriney, 2022; Laghaei et al., 2018; Meriney & Dittrich, 2013). Indeed, previously it has been shown that there is significantly reduced P/Q type VGCC labelling at aged mouse NMJs (Nishimune et al., 2016). Therefore, our observation of enhanced

tetanic potentiation could be a consequence of reduced $Ca^{2+}$ influx into each AZ during action potential activity, which would lead to a reduced probability of release. It is also possible that there is a homeostatic change in $Ca^{2+}$ dynamics or the mechanisms by which $Ca^{2+}$ triggers vesicle fusion, such that there is enhanced residual $Ca^{2+}$ after each action potential stimulation that leads to enhanced tetanic potentiation only among weaker synapses at 26 months. Interestingly, we did not detect tetanic potentiation in weak synapses at 4 months. We hypothesise that, at 4 months old, NMJs with low QC are on thin muscle fibres, and NMJs onto thin fibres scale their size as they require fewer quanta to be released to reach threshold (Katz & Thesleff, 1957). Under these conditions, AZs within these young synapses may have similar $Ca^{2+}$-dependent transmitter release mechanisms regardless of scaled NMJ size. Under these conditions, tetanic potentiation would not be observed in these 4-month-old NMJs with low QC. We also did not observe enhanced tetanic potentiation at later aged synapses (30 months). Perhaps the weaker synapses at 26 months that had altered AZ mechanisms may have been eliminated by 30 months, or at least the AZ mechanisms that led to tetanic potentiation do not exist any longer, resulting in a return to normal short-term synaptic plasticity. There remain unresolved questions behind the phenomena we observed in the early ageing phase, and it will be interesting to investigate potential presynaptic homeostatic plasticity mechanisms in the early ageing phase in future studies.

### Changes in RRP and VRR at aged neuromuscular synapses

RRP and VRR are important factors that contribute to the magnitude of transmitter release and the ability to maintain transmitter release during repeated use. The size of the RRP is proportional to the number of synaptic vesicles docked at AZs and this can be an important variable that controls the strength of a synapse (Dobrunz & Stevens, 1997; Waters & Smith, 2002). Once the RRP is depleted after several stimuli, the subsequent transmitter release is dependent on the vesicles from a 'recycling pool' replenishing the RRP (Elmqvist & Quastel, 1965; Richards et al., 2003). This rate of replenishment of the RRP is termed the VRR (Birks & Macintosh, 1961; Larkman et al., 1991). Our data showed that, although both RRP and VRR are correlated with QC (Fig. 4*E–J*), they appear to be modulated independently. We found a significant and robust reduction in VRR at NMJs from 30-month-old mice compared to NMJs from 4- or 26-month-old mice, but only a small but significant reduction in RRP at NMJs from 30-month-old mice

compared to NMJs in 26-month-old mice. Such changes could be caused by the reduction in the density of AZs we observed only at later ageing synapses (30 months). Prior studies have demonstrated independent regulation of these two properties during presynaptic homeostatic plasticity, which resulted in an acute change in RRP without a change in VRR (Wang et al., 2016). There are also a number of studies reporting that the RRP size is plastic and is responsive to postsynaptic perturbation, possibly as a homeostatic response (Goel et al., 2017; Kiragasi et al., 2017; Muller & Davis, 2012; Wang et al., 2016; Weyhersmuller et al., 2011). Furthermore, genetic studies eliminating RIM (Deng et al., 2011; Han et al., 2011; Kaeser et al., 2011; Muller et al., 2012) or Munc13 (Andrews-Zwilling et al., 2006; Deng et al., 2011; Shin et al., 2010) have been shown to impede RRP plasticity. The mechanisms that control the VRR may include presynaptic $Ca^{2+}$ (Sakaba, 2008) and cytomatrix proteins such as piccolo (Butola et al., 2017) and, because we detected a reduced $Ca^{2+}$ entry during the evoked events in 30–33-month-old NMJs, the reduced VRR that we observed could be a result of lower presynaptic $Ca^{2+}$ entry during action potential activity. More studies are needed to investigate the mechanisms by which these vesicle pool properties can be modulated at aged synapses.

### Reduced $Ca^{2+}$ entry during the evoked events at aged NMJs

Presynaptic $Ca^{2+}$ entry following action potential activity is a key regulating mechanism in the control of transmitter release. The reduction in presynaptic $Ca^{2+}$ that we observed in later aged synapses is consistent with the other age-induced presynaptic changes we report here and with a prior report of reduced immunohistochemical labelling of P/Q-type $Ca^{2+}$ channels at aged synapses (Nishimune et al., 2016). First, the reduced QC and EPP amplitude, with no change in mEPP amplitude and frequency, at 30-month-old NMJs could be explained by reduced presynaptic $Ca^{2+}$ entry because evoked release is dependent on presynaptic $Ca^{2+}$ influx, whereas spontaneous release is not (Fatt & Katz, 1952; Geppert et al., 1994). Second, the reduction that we report in AZ density could explain a reduced total $Ca^{2+}$ entry into NMJs, although it is also possible that there are changes in $Ca^{2+}$ channel density within each AZ.

Taken together, our data support the hypothesis that NMJs in the later ageing phase have reduced AZ density, which results in a reduction in VGCCs and a subsequent diminished presynaptic $Ca^{2+}$ entry during an action potential. The reduced presynaptic $Ca^{2+}$ entry could then lead to reduced QC and EPP amplitude and reduced VRR at aged NMJs.

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

## Additional information

### Data availability Statement

Data are available to qualified investigators by contacting the corresponding author.

## Competing interest

The authors declare that they have no competing interests.

## Author contributions

Y.L, Y.B and S.D.M were responsible for conceptualisation. Y.L and E.H.C were responsible for data acquisition. Y.L, C.B, A.M, M.G, E.H.C and A.J were responsible for data analysis. Y.L was responsible for writing the original draft. S.D.M and Y.B were responsible for reviewing and editing.

## Funding

Funding for this project was provided by NIH grants AG083078 and R21NS131752.

## Keywords

active zone, ageing, neuromuscular junction, neurotransmission, presynaptic release properties

## Supporting information

Additional supporting information can be found online in the Supporting Information section at the end of the HTML view of the article. Supporting information files available:

**Peer Review History**

