## [Peer Review History · The Journal of Physiology]

Aging-induced weakness of mouse NMJs is associated with reduced active zone density, synaptic event kinetics, and presynaptic calcium entry

Yizhi Li, Elinor H Case, Christopher Blanchard, Anna Monteleone, Meera Gandhi, Anousha Jaie, Yomna Badawi, and Stephen D Meriney
DOI: 10.1113/JP286735

Corresponding author(s): Stephen Meriney (meriney@pitt.edu)

The following individual(s) involved in review of this submission have agreed to reveal their identity: C. Andrew Frank (Referee #1)

Review Timeline:

Submission Date:	14-Mar-2025
Editorial Decision:	11-Apr-2025
Revision Received:	05-Jun-2025
Editorial Decision:	03-Jul-2025
Revision Received:	09-Jul-2025
Accepted:	22-Jul-2025

Senior Editor: Katalin Toth

Reviewing Editor: Samuel Young

Transaction Report:

Dear Dr Meriney,

Re: JP-RP-2025-286735 "**Aged-induced weakness of mouse NMJs is associated with reduced active zone density, vesicle replenishment rate, and synaptic event kinetics**" by Yizhi Li, Christopher Blanchard, Anna Monteleone, Meera Gandhi, Anousha Jaie, Yomna Badawi, and Stephen D Meriney

Thank you for submitting your manuscript to The Journal of Physiology. It has been assessed by a Reviewing Editor and by 2 expert referees and we are pleased to tell you that it is acceptable for publication following satisfactory revision.

REVISION CHECKLIST:

We look forward to receiving your revised submission.

Yours sincerely,

Katalin Toth
Senior Editor
The Journal of Physiology

REQUIRED ITEMS

- Author photo and profile. First or joint first authors are asked to provide a short biography (no more than 100 words for one author or 150 words in total for joint first authors) and a portrait photograph. These should be uploaded and clearly labelled together in a Word document with the revised version of the manuscript. See Information for Authors for further details.

- You must start the Methods section with a paragraph headed Ethical approval (https://jp.msubmit.net/cgi-bin/main.plex?form_type=display_requirements#methods).

Research must comply with The Journal's policies regarding animal experiments (<https://physoc.onlinelibrary.wiley.com/hub/animal-experiments>) and adherence to these policies must be stated in the manuscript.

Authors should confirm in their Methods section that their experiments were carried out according to the guidelines laid down by their institution's animal welfare committee, including an ethics approval reference number. The Methods section must contain a statement about access to food, water and housing, details of the anaesthetic regime: anaesthetic used, dose and route of administration, and method of killing the experimental animals.

- Please upload separate high-quality figure files via the submission form.

- Please ensure that the Article File you upload is a Word file.

- Your paper contains Supporting Information of a type that we no longer publish, including supplementary tables and figures. Any information essential to an understanding of the paper must be included as part of the main manuscript and figures. The only Supporting Information that we publish are video and audio, 3D structures, program codes and large data files. Your revised paper will be returned to you if it does not adhere to our Supporting Information Guidelines.

- Papers must comply with the Statistics Policy: https://jp.msubmit.net/cgi-bin/main.plex?form_type=display_requirements#statistics.

In summary:

- If $n \leq 30$, all data points must be plotted in the figure in a way that reveals their range and distribution. A bar graph with data points overlaid, a box and whisker plot or a violin plot (preferably with data points included) are acceptable formats.

- If $n > 30$, then the entire raw dataset must be made available either as supporting information, or hosted on a not-for-profit repository, e.g. FigShare, with access details provided in the manuscript.

- 'n' clearly defined (e.g. x cells from y slices in z animals) in the Methods. Authors should be mindful of pseudoreplication.
- All relevant 'n' values must be clearly stated in the main text, figures and tables.
- The most appropriate summary statistic (e.g. mean or median and standard deviation) must be used. Standard Error of the Mean (SEM) alone is not permitted.
- Exact p values must be stated. Authors must not use 'greater than' or 'less than'. Exact p values must be stated to three significant figures even when 'no statistical significance' is claimed.

- Please include an Abstract Figure file, as well as the Figure Legend text within the main article file. The Abstract Figure is a piece of artwork designed to give readers an immediate understanding of the research and should summarise the main conclusions. If possible, the image should be easily 'readable' from left to right or top to bottom. It should show the physiological relevance of the manuscript so readers can assess the importance and content of its findings. Abstract Figures should not merely recapitulate other figures in the manuscript. Please try to keep the diagram as simple as possible and without superfluous information that may distract from the main conclusion(s). Abstract Figures must be provided by authors no later than the revised manuscript stage and should be uploaded as a separate file during online submission labelled as File Type 'Abstract Figure'. Please also ensure that you include the figure legend in the main article file. All Abstract Figures should be created using BioRender. Authors should use The Journal's premium BioRender account to export high-resolution images. Details on how to use and access the premium account are included as part of this email.

EDITOR COMMENTS

Reviewing Editor:

Comments for Authors to ensure the paper complies with the Statistics Policy (Required):
Authors need to report data as SD, not SEM

Comments to the Author (Required):

This manuscript examines the relationship between synaptic function and age at the mouse NMJ. Both reviewers found the data to be impactful, important, and supported the conclusions drawn. Both reviewers has specific concerns about the clarity of the conclusions drawn. Therefore, the authors need to carefully revise and rewrite their manuscript and provide more text in the discussion given the positive and careful comments by the reviewers. In addition, to comply with Journal policy, the authors must report data as SD, not SEM.

Please also see 'Required Items' above.

REFEREE COMMENTS

Referee #1:

The manuscript by Li et al. examines potential mechanisms to explain weakened NMJ synapses in aging animals. The current work is a close companion of the Li et al., 2023 J. Physiology publication, and it is important to examine the manuscripts together.

For the prior study, the authors documented age-related dynamics in mouse ETA muscle NMJ function (Li 2023, Fig. 1). Specifically, they found small but significant increases in EPP amplitude and quantal content as the animals aged from early adulthood (3-6 months) to adulthood (7-24 months), but then the NMJs experienced a sudden decline in function at advanced ages (25-30 months). There was an earlier decline in gross motor axon innervation preceding the functional decline (~16 months; 2023, Fig. 2). And short-term facilitation responses were smaller early (2023, Fig. 4) before increasing. That prior study finished by focusing on the animals where NMJs have been weakened with extreme age. For those NMJs, there was improvement in function when a calcium channel potentiator GV-58 was added (2023, Fig. 5). That suggests that the problem at older mouse NMJs may be fixable, in some manner.

The current study focuses on those older NMJs and the mechanisms underlying the late-aged synapse phenotypes. The data compare NMJ properties from young adults (4 months of age) to aged animals (26 months) to extreme aged animals, near the end of a lab mouse lifespan (30 months). Importantly, instead of putting animals aged 25-30 months all in the same group, they are delineated into finer age ranges, which turns out to offer better resolution.

The data are interesting. The authors confirm the core neurotransmission findings of Li et al., 2023. But importantly, they find a steep drop-off in electrophysiological parameters between 26 and 30 months of age (Fig. 1). This drop-off in function correlates with a drop-off in active zone density, as assessed by Piccolo and RIMBP2 staining (Fig. 2). Consistently, the RRP size and vesicle replenishment rate (VRR) are diminished at 30 months of age. Finally, the prior finding that short-term facilitation responses are smaller early (4 months) than late (26 months) still holds (Fig. 3). But by 30 months of age, those values slip back down to a level similar to those measured at 4 months of age.

The results of the study support the conclusions. The dataset is a welcome companion to the prior findings because it offers new insights. The current study falls a little short of full mechanistic understanding, but that is not necessarily the applicable threshold. This reviewer has comments about specific points and puzzles, and most are (likely) addressable without new data.

Specific Points

1. The most interesting puzzle for this reviewer is trying to untangle the temporal sequence of events at the NMJ - i.e., what happens from young to old and what all the phenotypes described in the two manuscripts collectively mean.

Just examining this study, there is a good correlation between the most severe visual phenotypes (e.g., active zone density at 30 months) and the most severe functional phenotypes (e.g., EPP amplitude at 30 months). But taking the two studies together, there is a bit of a mismatch in the timing of when synaptic phenotypes pop up. For example, the innervation profile decreases well before functional output drops. Maybe this developmental decrease is a pre-ageing phenotype, or maybe it is unrelated.

This reviewer is hoping that the authors could offer Discussion material (or data from elsewhere) that addresses this puzzle. What happens when, and ultimately, what could it mean for function?

2. There is a sharp line where severe phenotypes develop at some point between 26 and 30 months of age. This is very interesting in terms of an ageing model. The authors suggest that the functional phenotypes may be a result of the diminished active zone material, which makes sense. They also offer that the electrophysiological kinetics phenotypes (e.g., rise time and decay time) might be a result of mis-apposition of pre- and postsynaptic components.

It might be possible to test this latter idea without new experiments and newly aged animals. The prior study (Li et al., 2023) has nice GV-58 data to show that it was possible to fix functional phenotypes at defective synapses, simply by adding a calcium channel potentiator. Could the authors retrospectively analyze the kinetic data from the GV-58 recordings? This way they could test if the kinetics were fixed there too, in addition to the EPPs. If the kinetics were fixed, then mis-apposition is likely not the explanation because the synaptic components would still be misaligned, GV-58 or not.

3. Regarding the mis-apposition idea, there is a confocal imaging experiment in this study (Fig. 2). Were these samples separately stained for BTX, or were they triply stained with BTX, Piccolo, and RIMBP2? The reviewer is wondering if it is possible to get information from 30-month-old mouse NMJs regarding the apposition of pre- and postsynaptic components, from images that already exist.

4. The short-term plasticity experiments are helpful to understand the NMJ's capacity for acute change in response to a challenge like a train. But it is not entirely clear to this reviewer what all the data mean and why the values at 30 months slip back down to a similar value as at 4 months. Between the two papers, the authors have found:

4-6 months - "low" short-term facilitation

7-26 months - "high" short-term facilitation

30 months - "low" short-term facilitation

The authors propose that there is a subset of low QC range synapses at 26 months that are responsible for all the short-term synaptic facilitation. That could be true. But if it were, would it also imply that all along (from 7-26 months) there is always a subset of low QC range synapses?

And why are the low QC synapses in the young animals (4 months, Fig. 3B) resistant to short-term facilitation? Could it be that the 4-month-old NMJs have not acquired this plastic capacity yet (because they have not set aside a cache of low QC sites yet), and the 30-month-old NMJs have lost the plastic capacity by eliminating them? Any Discussion insights or data cited from elsewhere would be welcome.

Referee #2:

This manuscript addresses the effect of aging on NMJ function in mouse. The manuscript compares synaptic amplitudes, kinetics and release properties. A principal finding that AZ density correlates to loss of function with age is important. I have a number of specific questions about the work and its discussion.

Main points

- 1) The authors focus on properties of synaptic release but perhaps consideration should also be made to postsynaptic effects. I note that in aged animals the mEPP amplitude is unchanged, but so is the frequency. This does not lend to any particular conclusion that isolates response alterations to a particular site. Some discussion in how postsynaptic changes in receptor number or alignment may alter conclusions is warranted.
- 2) In figure 2 comparisons were made between repetitive stimulation amplitudes in low medium and high QC terminals across age. The authors conclude that at 26 months low intensity synapses only, show increases in tetanic potentiation. Another possibility is that the types of synapses are fluid over time, and tetanic potentiation compensates for weaknesses as they occur.
- 3) The treatment and discussion of quantal content is difficult to follow. On the one hand QC is related to changes in rate (risetimes and decays) of minis (perhaps also evoked events). Citing earlier work from the senior author's group in the discussion it is suggested here that a change in QC could result from a misaligned set of presynaptic structures. Earlier work may indicate a change in Calcium channel coupling to the fusion machinery and here it is proposed that this could be an effect on laminins. I cannot see how this would relate to a change in mEPP kinetics. Some clarity in the discussion would help me.
- 4) Presynaptic Calcium influx may play a strong role and the authors note this in the conclusion. I understand it would be a lot of work but gaining some data on evoked Calcium transients in different synapse QC types at different ages would answer a lot of questions.

Minor points

- 1) In a number of cases raw data (or examples) are not shown or shown but not in a way that clarifies the point being made

For example in fig 1 evoked events are shown but the risetimes and decay rates are visible only in the graphs. Why not overlay mean responses scaled to the peak. Raw data for mEPPs is not really visible.

Raw data for fig 3 can be assumed to be in figure 4

2) The figure legends are very heavy in data points repeated from the graphs in the figures. This makes reading the figures and results challenging. If the authors or editor feel that this detail is necessary, results could be placed in a table or in the main text.

3) Wording of conclusions can be difficult to follow. For example the authors' state "Our results suggest that AZ density is potentially a major contributor to age-related reductions in neurotransmission. In addition, we found no correlation between AZ density and quantal content (Fig. 2I), which leads us to conclude that differences in synaptic strength may not have a lower AZ density, but may simply have differences in NMJ size and AZ number." I understand the point, but this is a convoluted way of making the statement.

END OF COMMENTS

We thank the reviewers for their careful evaluation of our manuscript. Below, we outlined each comment and our response. In addition, we have edited the manuscript as outlined below and believe the revised manuscript has been improved as a result of these changes.

Referee #1:

1. The most interesting puzzle for this reviewer is trying to untangle the temporal sequence of events at the NMJ - i.e., what happens from young to old and what all the phenotypes described in the two manuscripts collectively mean.

This reviewer is hoping that the authors could offer Discussion material (or data from elsewhere) that addresses this puzzle. What happens when, and ultimately, what could it mean for function?

We agree with this suggestion from the reviewer. We believe that the current study both confirmed and extended our understanding of the temporal sequence in NMJ aging: the synapses go through an initial phase where postsynaptic receptors break into islands, an early aging phase where potential presynaptic homeostatic compensation takes place, and then a later aging phase where additional structural deficits manifest and functional deficits in neurotransmission become prominent. We have confirmed in this study some of the structural changes, such as reduced AZ density, accompany the reduction in neurotransmission. This gives us a biologically based understanding of the aging time course. We have updated our manuscript (page 26) to present our summary of these changes.

2. There is a sharp line where severe phenotypes develop at some point between 26 and 30 months of age. This is very interesting in terms of an ageing model. The authors suggest that the functional phenotypes may be a result of the diminished active zone material, which makes sense. They also offer that the electrophysiological kinetics phenotypes (e.g., rise time and decay time) might be a result of mis-apposition of pre- and postsynaptic components. Could the authors retrospectively analyze the kinetic data from the GV-58 recordings? This way they could test if the kinetics were fixed there too, in addition to the EPPs. If the kinetics were fixed, then mis-apposition is likely not the explanation because the synaptic components would still be misaligned, GV-58 or not.

We agree that future research should be devoted to an investigation into the possibility of synaptic misalignment. However, we cannot use the kinetic data from GV-58 recordings because we have shown that GV-58 slows calcium channel mean open time, and thus also slows synaptic kinetics after acute application (Fig. 3B in Tarr et al., 2013. J Neurosci 33: 10559). It would be hard to separate age-induced changes in synaptic kinetics from GV-58 induced changes. We believe a future electronic microscopic study

on the synaptic alignment of aged NMJ would be the best method to address this question.

3. Regarding the mis-apposition idea, there is a confocal imaging experiment in this study (Fig. 2). Were these samples separately stained for BTX, or were they triply stained with BTX, Piccolo, and RIMBP2? The reviewer is wondering if it is possible to get information from 30-month-old mouse NMJs regarding the apposition of pre- and postsynaptic components, from images that already exist.

Yes, our samples were triple stained with BTX, Piccolo, and RIMBP2 and we agree that it would be very useful to extract data regarding synaptic apposition from our existing images. However, our confocal imaging data does not provide the resolution needed to get information about the apposition of pre- and postsynaptic components. We believe that future electron microscopic studies would be best suited to study the potential mis-apposition.

4. The short-term plasticity experiments are helpful to understand the NMJ's capacity for acute change in response to a challenge like a train. But it is not entirely clear to this reviewer what all the data mean and why the values at 30 months slip back down to a similar value as at 4 months. The authors propose that there is a subset of low QC range synapses at 26 months that are responsible for all the short-term synaptic facilitation. That could be true. But if it were, would it also imply that all along (from 7-26 months) there is always a subset of low QC range synapses? And why are the low QC synapses in the young animals (4 months, Fig. 3B) resistant to short-term facilitation? Could it be that the 4-month-old NMJs have not acquired this plastic capacity yet (because they have not set aside a cache of low QC sites yet), and the 30-month-old NMJs have lost the plastic capacity by eliminating them? Any Discussion insights or data cited from elsewhere would be welcome.

We thank the reviewer for these thought-provoking questions as we agree that the underlying mechanisms here are of interest. Of course, there is a subset of low quantal content NMJs in 4 and 26-month-old mice, however, the mechanisms behind weakness at different ages could be different: we hypothesize that low QC NMJs at 4 months of age represent a group of synapses that are on thin muscle fibers and that NMJs scale to muscle fiber size but AZs within these synapses are very similar to those onto larger diameter muscle fibers with high QC. At 26 months, as age-induced changes are occurring that compromise synaptic structure and function, we hypothesize that there are compensatory mechanisms in weaker NMJs that alter the AZs resulting in enhanced tetanic potentiation. We hypothesize that at 30 months of age, these weaker synapses that struggled previously may have been eliminated, or at least the AZ mechanisms that led to tetanic potentiation do not exist any longer, resulting in a return to normal short-

term synaptic plasticity. We have edited our discussion to better address these issues (Pages 28-29).

Referee #2:

1) The authors focus on properties of synaptic release but perhaps consideration should also be made to postsynaptic effects. I note that in aged animals the mEPP amplitude is unchanged, but so is the frequency. This does not lend to any particular conclusion that isolates response alterations to a particular site. Some discussion in how postsynaptic changes in receptor number or alignment may alter conclusions is warranted.

While our data lead us to hypothesize a potential synaptic misalignment, we agree that there could be post synaptic receptor mediated changes. We have added text on page 27 to acknowledge the potential involvement of post synaptic receptor function, density and/or distribution.

2) In figure 2 comparisons were made between repetitive stimulation amplitudes in low medium and high QC terminals across age. The authors conclude that at 26 months low intensity synapses only, show increases in tetanic potentiation. Another possibility is that the types of synapses are fluid over time, and tetanic potentiation compensates for weaknesses as they occur.

We agree with the reviewer that it is possible that AZ synaptic mechanisms may be fluid over time and we have revised the manuscript to present a variety of potential mechanisms (Pages 28-29).

3) The treatment and discussion of quantal content is difficult to follow. One the one hand QC is related to changes in rate (risetimes and decays) of minis (perhaps also evoked events). Citing earlier work from the senior author's group in the discussion it is suggested here that a change in QC could result from a misaligned set of presynaptic structures. Earlier work may indicate a change in Calcium channel coupling to the fusion machinery and here it is proposed that this could be an effect on laminins. I cannot see how this would relate to a change in mEPP kinetics. Some clarity in the discussion would help me.

We have revised and expanded our discussion of these issues in the revised document. First, a discussion of the older paper that focused on a low calcium-induced disruption of AZs and subsequent effects on quantal content (Meriney et al., 1996) is not closely related to our observations here, resulting in confusion for the reader, and has been removed. With regard the changes in mEPP kinetics, we propose that this might only occur if there were a misalignment between AZs that release mEPPs and the

postsynaptic receptor folds that concentrate receptors (pages 26-27). Such misalignment has been observed with laminin alpha 4 experimental knockout, and in aging a reduction in laminin alpha 4 has been previously reported. As such, we hypothesize that the increase in diffusion distance between release sites and receptors might create the alter mEPP kinetics. This hypothesis will have to be explored in future studies.

4) Presynaptic Calcium influx may play a strong role and the authors note this in the conclusion. I understand it would be a lot of work but gaining some data on evoked Calcium transients in different synapse QC types at different ages would answer a lot of questions.

We agree with the reviewer and have added calcium imaging data as requested to compare presynaptic calcium entry following a single action potential between young and aged NMJs. It was not feasible to do this for matched synapses for which we had QC data, but we feel that the sample we collected would cover a range of QC types. We found that aged NMJs showed significant less presynaptic calcium entry following a single action potential. These data are presented in a new figure (Fig. 5), with the methods given on pages 9-10 , the results presented on page 14, and the discussion presented on page 30.

Minor points

1) In a number of cases raw data (or examples) are not shown or shown but not in a way that clarifies the point being made. For example in fig 1 evoked events are shown but the risetimes and decay rates are visible only in the graphs. Why not overlay mean responses scaled to the peak. Raw data for mEPPs is not really visible. Raw data for fig 3 can be assumed to be in figure 4

We agree with the reviewer that showing additional raw data would illustrate our findings more clearly. We have added additional raw data to figures 1 and 3.

2) The figure legends are very heavy in data points repeated from the graphs in the figures. This makes reading the figures and results challenging. If the authors or editor feel that this detail is necessary, results could be placed in a table or in the main text.

The journal requires that we present p values for all the statistical comparisons made in our dataset, although we agree with the reviewer that this creates some difficulty reading the legends. We have edited the legends in an attempt to make them easier to read.

3) Wording of conclusions can be difficult to follow. For example the authors' state "Our results suggest that AZ density is potentially a major contributor to age-related reductions in neurotransmission. In addition, we found no correlation between AZ density and quantal content (Fig. 2I), which leads us to conclude that differences in synaptic strength may not have a lower AZ density, but may simply have differences in NMJ size and AZ number." I understand the point, but this is a convoluted way of making the statement.

We agree with the reviewer that this section of the manuscript was not worded clearly, and we have revised this text to make the presentation clearer (pages 27-28).

Dear Dr Meriney,

Re: JP-RP-2025-286735R1 "**Aging-induced weakness of mouse NMJs is associated with reduced active zone density, synaptic event kinetics, and presynaptic calcium entry**" by Yizhi Li, Elinor H Case, Christopher Blanchard, Anna Monteleone, Meera Gandhi, Anousha Jaie, Yomna Badawi, and Stephen D Meriney

Thank you for submitting your manuscript to The Journal of Physiology. It has been assessed by a Reviewing Editor and by 2 expert referees and we are pleased to tell you that it is acceptable for publication following satisfactory revision.

REVISION CHECKLIST:

We look forward to receiving your revised submission.

Yours sincerely,

Katalin Toth
Senior Editor
The Journal of Physiology

EDITOR COMMENTS

Reviewing Editor:

The authors have done an excellent job of responding to previous reviewer's critiques and comments. For improved readability Figure 3 legend, please make a table that has all the statistics information.

REFEREE COMMENTS

Referee #1:

The authors have done a fine job addressing the prior round's comments by me (Reviewer #1) and by Reviewer #2. In response to Reviewer #2, the calcium imaging experiments that the authors added (New Figure 5) are nice demonstrations of reduced calcium entry at the aged NMJs. I also appreciate the nuances in the revised Discussion points.

Referee #2:

This manuscript is an updated version of a previously reviewed manuscript and represents an important analysis of the effects of aging at the neuromuscular junction.

In general my previous comments and those from referee 1 have been satisfactorily addressed.

However, the figure legend to figure 4 remains complex - very heavy in data points repeated from the graphs in the figure. I noted this earlier and I understand there are editorial policies. But as it stands the legend is not readable. In addition (and I failed to note this earlier) the authors do not specify the tests used against multiple data points from repeated stimulation. This should be done with ANOVAs considering the repeated comparisons and the means of testing should be clarified. I think the authors should separate p numbers into a table. Perhaps the authors can work with the editor to improve the presentation.

END OF COMMENTS

We thank the reviewers for their careful evaluation of our updated manuscript. We have edited the manuscript as outlined below and believe the revised manuscript has been improved as a result of these changes.

Referee #2:

1. Referee #2 emphasized their concern with readability of figure 3 legend. Additionally, referee #2 noted missing statistical testing method in the same figure. They suggest switching to a table format to make the figure legend more digestible.

We agree with this suggestion from the reviewer. We have added a table under figure legend 3, and the data with p values have been moved into the table. As for the statistical testing method, it was mentioned in the original figure legend, but perhaps due to poor readability, it was hidden to the readers. We hope with our updated table format, the information will be clearer and more accessible.

Dear Dr Meriney,

Re: JP-RP-2025-286735R2 "**Aging-induced weakness of mouse NMJs is associated with reduced active zone density, synaptic event kinetics, and presynaptic calcium entry**" by Yizhi Li, Elinor H Case, Christopher Blanchard, Anna Monteleone, Meera Gandhi, Anousha Jaie, Yomna Badawi, and Stephen D Meriney

We are pleased to tell you that your paper has been accepted for publication in The Journal of Physiology.

Yours sincerely,

Katalin Toth
Senior Editor
The Journal of Physiology

If you would like to receive our 'Research Roundup', a monthly newsletter highlighting the cutting-edge research published in The Physiological Society's family of journals (The Journal of Physiology, Experimental Physiology, Physiological Reports, The Journal of Nutritional Physiology and The Journal of Precision Medicine: Health and Disease), please click this link, fill in your name and email address and select 'Research Roundup':
<https://www.physoc.org/journals-and-media/membernews>

- You can help your research get the attention it deserves! Check out Wiley's free Promotion Guide for best-practice recommendations for promoting your work at: www.wileyauthors.com/eoo/guide. You can learn more about Wiley Editing Services which offers professional video, design, and writing services to create shareable video abstracts, infographics, conference posters, lay summaries, and research news stories for your research at: www.wileyauthors.com/eoo/promotion.

EDITOR COMMENTS

Reviewing Editor:

There are no further concerns.